# Analyzing Parameter-Efficient Convolutional Neural Network Architectures for Visual Classification

**DOI:** 10.3390/s25247663

**Published:** 2025-12-17

**Authors:** Nazmul Shahadat, Anthony S. Maida

**Affiliations:** 1School of Science and Mathematics, Truman State University, Kirksville, MO 63501, USA; 2School of Computing and Informatics, University of Louisiana at Lafayette, Lafayette, LA 70503, USA

**Keywords:** hypercomplex networks, quaternion networks, PHM layer, axial attention networks, representation learning, weight sharing, PHM based dense layer, deep learning, parameter efficient, cost effective, RCNs, residual axial networks

## Abstract

Advances in visual recognition have relied on increasingly deep and wide convolutional neural networks (CNNs), which often introduce substantial computational and memory costs. This review summarizes recent progress in parameter-efficient CNN design across three directions: hypercomplex representations with cross-channel weight sharing, axial attention mechanisms, and real-valued architectures using separable convolutions. We highlight how these approaches reduce parameter counts while maintaining or improving accuracy. We further analyze our contributions within this landscape. Full hypercomplex neural networks (FHNNs) employ hypercomplex layers throughout the architecture to reduce latency and parameters, while representational axial attention models (RepAA) extend this efficiency by generating additional feature representations. To mitigate the remaining overhead of spatial hypercomplex operations, we introduce separable hypercomplex networks (SHNNs), which factorize quaternion convolutions into sequential vectormap operations, lowering parameters by approximately 50%. Finally, we compare these models with popular efficient architectures, such as MobileNets and SqueezeNets, and demonstrate that our residual one-dimensional convolutional networks (RCNs) achieve competitive performance in image classification and super-resolution with significantly fewer parameters. This review highlights emerging strategies for reducing computational overhead in CNNs and outlines directions for future research.

## 1. Introduction

All things being equal, deeper or wider convolutional neural networks (CNNs) perform better than shallower or thinner networks [1]. This statement encourages the research community to explore state-of-the-art deeper CNNs and utilize more channels. These deeper and wider networks improve performance on image classification and recognition, speech recognition, pattern recognition, prediction of gene mutations, natural language processing and sentiment analysis, and more [2].

However, these networks tend to have higher computational costs. These costs increase linearly with network depth; for example, the 26, 35, 50, 101, and 152 layer residual networks require 41M, 58M, 83M, 149M, and 204M parameters, and 2.6G, 3.3G, 4.6G, 8.8G, and 13.1G FLOPs, respectively [3,4,5]. Moreover, the costs of wider networks increase exponentially with the widening factor; for example, the 26-layer WRNs with widening factors 1, 2, 4, 6, 8, and 10 are trained with 41M, 163M, 653M, 1469M, 2612M, and 4082M parameters and 2.5G, 10G, 41G, 92G, 163G, and 255G FLOPs, respectively [1,6].

We introduce novel CNNs that achieve near state-of-the-art visual classification performance, or better, while using fewer parameters. We call this topic *parameter efficiency*. One approach toward parameter efficiency is to broaden the types of weight sharing used in the network by introducing hypercomplex numbers. Another approach involves adjusting the architecture to use fewer parameters, such as axial or separable designs. For one example, weight sharing across space allows CNNs to drastically reduce trainable counts while improving image classification accuracy as compared to classical fully connected networks for visual classification and object detection tasks [7,8,9,10], image segmentation [11,12] and captioning [13]. Standard CNNs exploit weight sharing across spatial feature maps (height and width) but do not share weights across channels.

While developing hypercomplex CNNs (HCNNs), it was found that they share weights across channels as well as space [4,7]. This channel-based weight sharing can further reduce parameter requirements over standard CNNs. Their design is based on complex or hypercomplex number systems, usually with dimensions of two or four (but they can also be eight or sixteen dimensions using the octonion number system [14] or sedenion number system, respectively). HCNN operations with different dimensions can be used for real-world applications using the complex number system (2D) [15,16], or the quaternion number system (4D) [8,9]. Also, it was found that the CNN calculations in an HCNN only use a subset of the number system. This removes the dimensionality constraint and allows the calculations to be replicated for any dimensions [4,17]. The success of all these HCNNs appears to be due to their cross-channel weight-sharing mechanism and consequent ability to treat data across feature maps coherently [8,9]. This space- and channel-based weight sharing reduces parameter costs significantly.

However, these early HCNNs did not use hypercomplex operations in all layers. Their fully connected backends were still real-valued. Full hypercomplex networks (FHNNs), introduced in [5], include hypercomplex calculations in all layers, including the fully connected backend. These FHNNs reduce parameters moderately.

A separate concept, the axial concept was also used to reduce costs [11,18]. This concept splits the CNN operation into separate consecutive operations on the height and width input axes for vision tasks. These axial operations reduce costs O(h+w)=O(2h) from $O\left(hw\right)=O\left(h^{2}\right)$ where *h* and *w* are the height and width of a 2D image. We review this axial concept for attention-based autoregressive models [11,18] and CNNs [19].

Another CNN-based parameter-efficient concept has been introduced to split a k×k convolutional filter into two filters k×1 and 1×k [12]. This separable CNN reduces cost from $O(k^{2})$ to O(2k). Also the channel reduction concept has been proposed to reduce parameters, although it also reduces the model’s performance [10,20,21]. We review these concepts and trade-offs and compare them with the residual 1D convolutional networks (RCNs). The current paper analyzes separable FHNNs proposed by [22] to analyze more parameter-efficient HCNNs.

This paper first reviews parameter efficiency for recent work and develops new CNN models in real-valued and hypercomplex space. It analyzes ways to reduce computational costs for CNNs in vision tasks. We review the parameter efficiency properties of HCNNs and then introduce several novel hypercomplex works that use weight sharing across channels and space. In modern deep CNNs, many convolutional layers are stacked, including the frontend (stem) layers, the backend layers, and the network blocks. This paper introduces and studies full hypercomplex networks (FHNNs) where hypercomplex dense layers are used in the backend of a hypercomplex network. It helps to reduce costs in the densely connected backend. This paper also studies attention-based CNNs, where feature maps generated by an HCNN layer are used in attention-based autoregressive models, and channel-based weight sharing is used. This channel-based weight sharing helps to reduce costs for attention-based networks, and the HCNN feature maps for the attention module help increase performance. The parameter efficiency for FHNNs can, however, be improved further. Separable HCNNs (SHNNs) have been introduced for vision classification [22] to make CNNs more parameter efficient. SHNNs outperform the other HCNNs with lower costs (parameters, FLOPs, and latency).

We then study parameter efficiency for real-valued CNNs. These CNNs extract local features and construct residual structures with identity mapping across layers [3], MobileNets [10,21], SqueezeNets [12,20], and image super-resolution architectures [23,24,25,26]. As images are two-dimensional, these standard CNNs consume $O(h^{2}k^{2})$ computational costs (height *h*, and width *w*) per CNN layer. This paper analyzes a novel structure with 1D CNN (Conv1D) for vision to replace standard 2D CNN (Conv2D) and to reduce costs [19]. The Conv1D consumes O(hk) compared to $O(h^{2}k^{2})$ of Conv2D. As two consecutive Conv1D are used instead of a Conv2D, a pair of layers costs O(2hk). A residual connection is used in each Conv1D layer dealing with gradient problems when training deep networks and improving the performance of network architectures [19]. We also studied related work to compare it with this novel network. Our mathematical analysis shows the parameter efficiency, and empirical analysis shows that this novel real-valued CNN outperformed with fewer parameters, FLOPs, and latency.

## 2. Goals and Contributions

This paper analyzes the parameter efficiency of widely used CNN architectures, including hypercomplex CNNs, attention-based CNNs, and standard real-valued CNNs (for both computer and mobile settings), on visual classification tasks. This study offers valuable insights for designing cost-effective and parameter-efficient models. Prior hypercomplex CNNs leverage cross-channel weight sharing via the Hamiltonian product to reduce parameters, offering a strong foundation for efficiency [2,4,9]. However, these models still use a conventional real-valued fully connected backend. We extend hypercomplex operations throughout the network (including the dense backend using a Parameterized Hypercomplex Multiplication layer) to construct a fully hypercomplex model. This design improves classification accuracy while reducing the parameter count and FLOPs compared to related networks, confirming the benefits of end-to-end hypercomplex modeling [2]. Nevertheless, the reduction in parameters and computation from this step alone is modest. To significantly further improve efficiency, we introduce three novel architectures as outlined below.

### 2.1. Separable Hypercomplex Neural Networks (SHNN)

We propose Separable Hypercomplex Neural Networks (SHNN) to achieve greater parameter reduction in hypercomplex CNNs. In a SHNN, each 2D quaternion convolution (QCNN) (e.g., a 3×3 filter) is factorized into two sequential 1D convolutions (3×1 followed by 1×3) in the hypercomplex domain. Replacing a standard QCNN with two separable vectormap CNN layers yields a separable hypercomplex bottleneck block, which we stack to form the SHNN architecture. This separable design significantly reduces the number of parameters and FLOPs, while improving classification performance on CIFAR-10/100, SVHN, and Tiny ImageNet, compared to existing hypercomplex baselines. In fact, our SHNN models achieve state-of-the-art accuracy among hypercomplex networks. The only trade-off is a slight increase in latency. Overall, these results demonstrate that SHNN is a highly parameter-efficient hypercomplex CNN (HCNN) design.

### 2.2. Representational Axial Attention (RepAA) Networks

We introduce Representational Axial Attention (RepAA) networks to bring hypercomplex efficiency gains into attention-based models. Axial-attention CNNs are already more parameter efficient than full 2D attention models, yet they still rely on standard convolutional layers to compute attention weights. In our RepAA design, we incorporate HCNN layers with cross-channel weight sharing into the axial-attention architecture. Specifically, we replace key components of the axial-attention network (the early stem convolution, the axial attention blocks, and the final fully connected layer) with *representationally coherent* hypercomplex modules. The resulting RepAA network uses fewer parameters than the original axial-attention model while maintaining or improving classification accuracy. The result indicates that hypercomplex weight-sharing techniques can further reduce parameters in attention-based networks, making RepAA a promising approach for building more compact yet effective attention models.

### 2.3. Residual 1D Convolutional Networks (RCNs)

To improve efficiency in real-valued CNNs, we propose Residual 1D Convolutional Networks (RCNs), which serve as direct substitutes for 2D CNNs in networks like ResNet. In an RCN block, a 2D spatial convolution is decomposed into two sequential 1D convolutions with an interleaved residual connection. We integrated RCN blocks into various ResNet architectures and found that RCN-based ResNets achieved better accuracy while using about 80% fewer parameters, 30% fewer FLOPs, and 35% lower inference latency. When we applied the RCN concept to other models, we observed significant reductions in computational cost with performance gains. These results confirm the effectiveness of the RCN architecture for creating lightweight, high-performing real-valued CNNs.

## 3. Background and Related Work

This section reviews recent deep networks designed to reduce the parameters, FLOPs, and latency for vision tasks.

### 3.1. Standard Convolutional Neural Networks

Deep CNN architectures for vision consist of multiple layers that perform operations such as convolution, batch normalization, and non-linear activations. The trainable CNN step is crucial for extracting features from the input to produce the output.

#### 3.1.1. The Convolution Operation

The convolution operation in CNNs is simple yet versatile, applied throughout various network stages. It draws inspiration from the mammalian visual cortex, where individual cortical neurons respond to stimuli within a limited area called the receptive field. These receptive fields overlap and collectively cover the entire visual field.

The convolution operation is fundamental to convolutional models, using small neighborhoods defined by the kernel sizes to learn pixel correlations in the input space. For an input image $X\in R^{h\times w\times d_{in}}$ with height *h*, width *w*, and input channel count $d_{in}$, the convolution operation operates in a square local neighborhood $N_{k(i,j)=\{a,b||a-i|\leq k/2,|b-j|\leq k/2\}}$ of input *X* centered on the pixel $X_{ij}$ with spatial extent *k* (*k* is odd and the kernel is square). This local neighborhood ($N_{k}$) extracts the region (a,b)∈X for the centered pixel (i,j) depicted in Figure 1a. These extracted input pixels operate by spatially summing the product of depthwise matrix multiplications with a learnable weight matrix $W\in R^{h\times w\times d_{out}\times d_{in}}$ shown in Figure 1b. The output $C_{ij}\in R^{d_{out}}$ for this operation in the neighborhood of $X_{ij}$ is defined as(1)$$C_{i,j,n}=\sum_{\left(a,b,m\right)\in N_{k\times k(i,j)}}W_{a,b,m,n}X_{i+a-1,j+b-1,m}$$
where *m* and *n* are the indices for the input channel $d_{in}$ and output channel $d_{out}$, and *W* is the shared weights to calculate the output for all pixel positions (i,j) [19].

Research in deep learning has increasingly focused on improving CNN architectures for various applications, including document reading, handwriting and speech recognition, image segmentation, self-driving cars, video analysis, natural language processing, object detection, and time series classification [27,28,29].

#### 3.1.2. Complexity Analysis of Standard CNNs

The standard convolution operation given in Equation (Equation 1) is parameterized by *k*, $d_{in}$, and $d_{out}$, whose computational cost can be calculated as ${\mathrm{Cost}}_{\mathrm{Conv}2D}\propto h\cdot w\cdot d_{in}\cdot d_{out}\cdot k\cdot k$, where the computational cost depends multiplicatively on the kernel size, the input feature map, and the input and output depths. For vision tasks, we assume the input image is square, so $h\cdot w=h^{2}$. So, the cost equation can be rewritten as follows:(2)$${\mathrm{Cost}}_{\mathrm{Conv}2D}\propto h^{2}\cdot d_{in}\cdot d_{out}\cdot k^{2}.$$

The complexity of spatial 2D CNN is $O(h^{2}d_{in}d_{out}k^{2})$, which, as we shall see, is more expensive than needed [19,21].

### 3.2. Hypercomplex Neural Networks

The standard CNN shares weights across space but not across channels. In contrast, the hypercomplex CNN (HCNN) shares weights across both channels and space, enabling the discovery of cross-channel correlations and producing superior output representations, such as reconstructing color from a grayscale image [18,30]. Danihelka et al. [31] demonstrate the benefits of complex-valued representations in associative memory.

The HCNN algebra can utilize various dimensional systems, including 2-dimensional (complex), 4-dimensional (quaternion), 8-dimensional (octonion), 16-dimensional (sedenion), and generalized hypercomplex systems [4,17]. These architectures facilitate weight sharing across space and channels, enabling the cohesive treatment of multi-dimensional data [8,18]. Notably, research has demonstrated that cross-channel weight sharing can help CNNs recover color from grayscale images, while standard CNNs handle color channels independently, hindering this reconstruction. This section covers several recently introduced parameter-efficient hypercomplex convolutional networks.

#### 3.2.1. Quaternion Convolutional Neural Networks

##### Background on Complex Networks

The complex number system, invented in 1833 [32], is used in deep networks for 2D inputs due to its capabilities for feature representation. The importance of complex numbers $i\equiv \sqrt{-1}$ is highlighted by Georgiou [33] in relation to quantum mechanics [34]. The concepts of the complex number system, perceptron learning rule, and backpropagation in this domain are discussed by Georgiou [33]. The properties of complex-valued neural networks (CVNNs) enable easier optimization, better generalization, faster learning, and noise-robust memory mechanisms [2,31]. Trabelsi et al. [15] establish methods to create viable deep CVNNs, including complex batch normalization and weight initialization. CVNNs have also been used in recurrent neural networks and other models to produce richer representations [2]. A complex number system is defined as z=x+iy, where *x* and *y* are the real numbers and *i* is $\sqrt{-1}$. In this equation, *x* is a real, and iy is an imaginary component. A complex filter matrix *F* is convolved with a input vector *M* where $F=F_{r}+iF_{i}$ and $M=M_{r}+iM_{i}$, we get,(3)$$M⊗F=\left(F_{r}*M_{r}-F_{i}*M_{i}\right)+i\left(F_{i}*M_{r}+F_{r}*M_{i}\right)$$
where $F_{r}$ and $F_{i}$ are real matrices, and $M_{r}$ and $M_{i}$ are real vectors.

##### Quaternion Convolutional Operation

The quaternion number system is four-dimensional compared to the 2D complex number system. Quaternion numbers are formulated as Q=r+ix+jy+kz where *r*, *x*, *y*, and *z* are real numbers and *i*, *j*, and *k* are distinct imaginary components. The quaternion and complex networks work almost the same way as convolution with the help of quaternion (QCNN), and complex filters, respectively. The QCNNs expand the CNN explained in Equation (Equation 3). The Hamilton product for two quaternions *M* (4D input vector, whose fields are $M^{1,1},M^{1,2},M^{2,1},M^{2,2}$) and *F* (2D filter matrix, whose fields are $F^{1,1},F^{1,2},F^{2,1},F^{2,2}$) is defined as(4)$$\begin{array}{cc} M⊗F & =M^{1,1}F^{1,1}+M^{1,2}F^{1,2}+M^{2,1}F^{2,1}+M^{2,2}F^{2,2}\\ \; & =\left(M_{r}^{1,1},M_{i}^{1,1},M_{j}^{1,1},M_{k}^{1,1}\right)\left(F_{r}^{1,1},F_{i}^{1,1},F_{j}^{1,1},F_{k}^{1,1}\right)\\ \; & +\left(M_{r}^{1,2},M_{i}^{1,2},M_{j}^{1,2},M_{k}^{1,2}\right)\left(F_{r}^{1,2},F_{i}^{1,2},F_{j}^{1,2},F_{k}^{1,2}\right)\\ \; & +\left(M_{r}^{2,1},M_{i}^{2,1},M_{j}^{2,1},M_{k}^{2,1}\right)\left(F_{r}^{2,1},F_{i}^{2,1},F_{j}^{2,1},F_{k}^{2,1}\right)\\ \; & +\left(M_{r}^{2,2},M_{i}^{2,2},M_{j}^{2,2},M_{k}^{2,2}\right)\left(F_{r}^{2,2},F_{i}^{2,2},F_{j}^{2,2},F_{k}^{2,2}\right) \end{array}$$
where ⊗ represents the Hamiltonian product [9], M⊗F, and all of the other symbols in Equation (Equation 4) are quaternion numbers. The first line of Equation (Equation 4) (RHS) defines the Hamiltonian product of *M* and *F*. A quaternion *F* can be represented as using the following matrix of real numbers [8,9],(5)$$\begin{array}{c} F=\left[\begin{array}{cccc} F_{r} & -F_{i} & -F_{j} & -F_{k}\\ F_{i} & F_{r} & -F_{k} & F_{j}\\ F_{j} & F_{k} & F_{r} & -F_{i}\\ F_{k} & -F_{j} & F_{i} & F_{r} \end{array}\right]. \end{array}$$

The above matrix representation is not unique, but we use the above matrix like Gaudet and Maida [9]. For a clearer explanation, we expand the first term $M^{1,1}F^{1,1}$ on the right side of Equation (Equation 4) by using the distributive property and grouping terms [9,35], defined as(6)$$\begin{array}{cc} M^{1,1}F^{1,1}=\left(O_{r},O_{i},O_{j},O_{k}\right)= & \left(M_{r}^{1,1},M_{i}^{1,1},M_{j}^{1,1},M_{k}^{1,1}\right)\left(F_{r}^{1,1},F_{i}^{1,1},F_{j}^{1,1},F_{k}^{1,1}\right)\\ = & (M_{r}^{1,1}F_{r}^{1,1}-M_{i}^{1,1}F_{i}^{1,1}-M_{j}^{1,1}F_{j}^{1,1}-M_{k}^{1,1}F_{k}^{1,1},\\ \; & M_{i}^{1,1}F_{r}^{1,1}+M_{r}^{1,1}F_{i}^{1,1}+M_{j}^{1,1}F_{k}^{1,1}-M_{k}^{1,1}F_{j}^{1,1},\\ \; & M_{j}^{1,1}F_{r}^{1,1}+M_{r}^{1,1}F_{j}^{1,1}+M_{k}^{1,1}F_{i}^{1,1}-M_{i}^{1,1}F_{k}^{1,1},\\ \; & M_{k}^{1,1}F_{r}^{1,1}+M_{r}^{1,1}F_{k}^{1,1}+M_{i}^{1,1}F_{j}^{1,1}-M_{j}^{1,1}F_{i}^{1,1}) \end{array}$$
where $\mathrm{re}\left(M^{1,1}F^{1,1}\right)=M_{r}^{1,1}F_{r}^{1,1}-M_{i}^{1,1}F_{i}^{1,1}-M_{j}^{1,1}F_{j}^{1,1}-M_{k}^{1,1}F_{k}^{1,1}$, $i\left(M^{1,1}F^{1,1}\right)=M_{i}^{1,1}F_{r}^{1,1}+M_{r}^{1,1}F_{i}^{1,1}+M_{j}^{1,1}F_{k}^{1,1}-M_{k}^{1,1}F_{j}^{1,1}$, $j\left(M^{1,1}F^{1,1}\right)=M_{j}^{1,1}F_{r}^{1,1}+M_{r}^{1,1}F_{j}^{1,1}+M_{k}^{1,1}F_{i}^{1,1}-M_{i}^{1,1}F_{k}^{1,1}$, and $k\left(M^{1,1}F^{1,1}\right)=M_{k}^{1,1}F_{r}^{1,1}+M_{r}^{1,1}F_{k}^{1,1}+M_{i}^{1,1}F_{j}^{1,1}-M_{j}^{1,1}F_{i}^{1,1}$. It means convolve the real-valued input channel with the filter channel to obtain a real-valued scalar. Hence, the real parts in Equations (Equation 4) and (Equation 5) equal $O_{r}$. Similarly, the other parts $O_{i}$, $O_{j}$, and $O_{k}$ are defined as(7)$$\begin{array}{cc} O_{r}= & \mathrm{re}\left(M^{1,1}F^{1,1}\right)+\mathrm{re}\left(M^{1,2}F^{1,2}\right)+\mathrm{re}\left(M^{2,1}F^{2,1}\right)+\mathrm{re}\left(M^{2,2}F^{2,2}\right)\\ O_{i}= & i\left(M^{1,1}F^{1,1}\right)+i\left(M^{1,2}F^{1,2}\right)+i\left(M^{2,1}F^{2,1}\right)+i\left(M^{2,2}F^{2,2}\right)\\ O_{j}= & j\left(M_{r}^{1,1}F_{r}^{1,1}\right)+j\left(M^{1,2}F^{1,2}\right)+j\left(M^{2,1}F^{2,1}\right)+j\left(M^{2,2}F^{2,2}\right)\\ O_{k}= & k\left(M^{1,1}F^{1,1}\right)+k\left(M^{1,2}F^{1,2}\right)+k\left(M^{2,1}F^{2,1}\right)+k\left(M^{2,2}F^{2,2}\right) \end{array}$$

The real component ($O_{r}$) of the convolution value is defined as(8)$$\begin{array}{cc} O_{r} & =M_{r}⊗F_{r}-M_{i}⊗F_{i}-M_{j}⊗F_{j}-M_{k}⊗F_{k}\\ \; & =M_{r}^{1,1}F_{r}^{1,1}+M_{r}^{1,2}F_{r}^{1,2}+M_{r}^{2,1}F_{r}^{2,1}+M_{r}^{2,2}F_{r}^{2,2}\\ \; & -M_{i}^{1,1}F_{i}^{1,1}-M_{i}^{1,2}F_{i}^{1,2}-M_{i}^{2,1}F_{i}^{2,1}-M_{i}^{2,2}F_{i}^{2,2}\\ \; & -M_{j}^{1,1}F_{j}^{1,1}-M_{j}^{1,2}F_{j}^{1,2}-M_{j}^{2,1}F_{j}^{2,1}-M_{j}^{2,2}F_{j}^{2,2}\\ \; & -M_{k}^{1,1}F_{k}^{1,1}-M_{k}^{1,2}F_{k}^{1,2}-M_{k}^{2,1}F_{k}^{2,1}-M_{k}^{2,2}F_{k}^{2,2} \end{array}$$

Similarly, we can define $O_{i}$, $O_{j}$, and $O_{k}$. Like Equations (Equation 5) and (Equation 6), they can be re-expressed as(9)$$\left[\begin{array}{c} R(M^{1,1}*F^{1,1})\\ I(M^{1,1}*F^{1,1})\\ J(M^{1,1}*F^{1,1})\\ K(M^{1,1}*F^{1,1}) \end{array}\right]=\left[\begin{array}{cccc} F_{r}^{1,1} & -F_{i}^{1,1} & -F_{j}^{1,1} & -F_{k}^{1,1}\\ F_{i}^{1,1} & F_{r}^{1,1} & -F_{k}^{1,1} & F_{j}^{1,1}\\ F_{j}^{1,1} & F_{k}^{1,1} & F_{r}^{1,1} & -F_{i}^{1,1}\\ F_{k}^{1,1} & -F_{j}^{1,1} & F_{i}^{1,1} & F_{r}^{1,1} \end{array}\right]*\left[\begin{array}{c} M_{r}^{1,1}\\ M_{i}^{1,1}\\ M_{j}^{1,1}\\ M_{k}^{1,1} \end{array}\right]$$

In Equation (Equation 9), each kernel channel is convolved with the corresponding image channel. Equations (Equation 4) and (Equation 6) show that each trainable filter parameter is used four times over the sixteen multiplications. Kernel reuse is how weight sharing occurs. The other three components $M^{1,2}F^{1,2},M^{2,1}F^{2,1},and\;M^{2,2}F^{2,2}$ in Equation (Equation 4) have the same structure, so the nature of the weight sharing is the same for all terms.

Quaternions enable networks to encapsulate four-dimensional input features, such as treating color as RGB plus grayscale. Quaternion neural networks (QNNs) were first introduced by Arena et al. [36] and are trained using backpropagation like traditional neural networks. These QNNs use quaternion-based loss functions, activation functions, batch normalization, and parameter initialization methods. Quaternions are applied in various areas, including 3D rotation and motion detection, document segmentation, auto denoising applications, 3D space transformations, and robot control [2,8].

##### Complexity Analysis

Equations (Equation 6) and (Equation 9) show how the filter $F^{1,1}$ is reused in four convolutions and how the weight sharing occurs across four input channels. So, the weight-sharing ratio is 1/4 for quaternions. By comparing with Equation (Equation 2), the cost of QCNNs is(10)$${\mathrm{Cost}}_{\mathrm{QCNN}}\propto \left(\frac{1}{4}\cdot h^{2}\cdot d_{in}\cdot d_{out}\cdot k^{2}\right)$$
where the computational cost reduces 1/4 times compared to the standard CNN, and the number of output channels (output depth). The complexity of QCNNs is $O(\frac{1}{4}h^{2}d_{in}d_{out}k^{2})$, which is still extensively expensive [4,5,8,9].

#### 3.2.2. Generalized Hypercomplex CNNs

As stated previously, HCNNs have dimensional restrictions to powers of two. This means that the cross-channel weight sharing is limited to 2, 4, 8, or 16 channels.

##### Vectormap CNNs

To remove these dimensionality restrictions, Gaudet and Maida [4] introduce vectormap CNNs (VCNNs). The weight-sharing ratio of a VCNN is $\frac{1}{N}$, where *N* is the dimension size. We re-expand the first term $M^{1,1}F^{1,1}$ on the right side of Equation (Equation 4) for dimension N=4 [4], defined as,(11)$$\left[\begin{array}{c} R(M^{1,1}*F^{1,1})\\ I(M^{1,1}*F^{1,1})\\ J(M^{1,1}*F^{1,1})\\ K(M^{1,1}*F^{1,1}) \end{array}\right]=L⊙\left[\begin{array}{cccc} F_{r}^{1,1} & F_{i}^{1,1} & F_{j}^{1,1} & F_{k}^{1,1}\\ F_{i}^{1,1} & F_{r}^{1,1} & F_{k}^{1,1} & F_{j}^{1,1}\\ F_{j}^{1,1} & F_{k}^{1,1} & F_{r}^{1,1} & F_{i}^{1,1}\\ F_{k}^{1,1} & F_{j}^{1,1} & F_{i}^{1,1} & F_{r}^{1,1} \end{array}\right]*\left[\begin{array}{c} M_{r}^{1,1}\\ M_{i}^{1,1}\\ M_{j}^{1,1}\\ M_{k}^{1,1} \end{array}\right]$$
where L is a learnable matrix $L\in R^{N\times N}.$ The initial matrix is formed using the following: (12)$$l_{ij}=\left\{\begin{array}{cc} 1 & i\;=\;1\\ 1 & i\;=\;j\\ 1 & j=Cal_{i}\;\mathrm{where}\;Cal_{i}=(i+\left(i-1\right))\;\;\&\\ \; & Cal_{i}=Cal_{i}-D_{vm}\;\mathrm{if}\;Cal_{i}>D_{vm}\\ -1 & \mathrm{else}. \end{array}\right.$$

For N=4, *L* is calculated using Equation (Equation 12) and is depicted as follows:(13)$$L_{4}=\left[\begin{array}{cccc} 1 & 1 & 1 & 1\\ -1 & 1 & 1 & -1\\ 1 & -1 & 1 & -1\\ -1 & -1 & 1 & 1 \end{array}\right]$$

Hence, we can derive a learnable matrix *L* for an arbitrary dimension. The weight and variable initialization of VCNNs follows He et al. [3], Gaudet and Maida [4].

##### Parameterized Hypercomplex Multiplication

Like VCNNs, parameterized hypercomplex multiplication (PHM) is another form of generalized hypercomplex networks used for fully connected (FC) layers [17]. The PHM can be any dimension given by *N* (same as in Section Vectormap CNNs). If the dimension is four (N=4), the PHM operation is similar to the Hamilton product of Equation (Equation 4). Since the PHM is designed for FC layers, our explanation is limited only to FC layers. An FC layer is defined as y=Wx+b, where $W\in R^{d_{out}\times d_{in}}$ and $b\in R^{d_{out}}$ are weights and bias, and $d_{in}$ and $d_{out}$ are input and output dimensions. However, the PHM-based FC layer transforms the input $x\in R^{d_{in}}$ into an output $y\in R^{d_{out}}$ as y=Hx+b, where H represents the PHM layer and $H\in R^{d_{out}\times d_{in}}$ which is generated from the sum of Kronecker products. Instead of initializing weight, the PHM operation generates the hypercomplex parameter $H\in R^{d_{out}\times d_{in}}$ from the sum of Kronecker product of parameter matrices $I_{i}\in R^{N\times N}$ and $A_{i}\in R^{d_{out}/N\times d_{in}/N}$, where i=1…N and *N* is the number of dimensions (N=4):(14)$$H=\sum_{i=1}^{N}I_{i}⊗A_{i}$$
where ⊗ denotes the Kronecker product (the same symbol is used in quaternion convolution in Section 3.2.1). The prerequisites of PHM layer operation are that the input dimensions $d_{in}$ and the output dimension $d_{out}$ are divisible by the number of dimensions *N*. The parameter reductions mainly come from reusing the parameter matrices *I* and *A* in the PHM layer. This paper explains the 4D PHM layer. The learnable parameter components for the four dimensions are $O_{r}$, $O_{i}$, $O_{j}$, and $O_{k}$, where *O* denotes the PHM layer with four dimensions:(15)$$O_{in}=O_{r}+O_{i}+O_{j}+O_{k}$$

The 4D hypercomplex parameter matrix can be defined (using Equation (Equation 9)) as follows:(16)$$H=\underset{I_{1}}{\underset{︸}{\left[\begin{array}{cccc} 1 & 0 & 0 & 0\\ 0 & 1 & 0 & 0\\ 0 & 0 & 1 & 0\\ 0 & 0 & 0 & 1 \end{array}\right]}}⊗\underset{A_{1}}{\underset{︸}{\left[\begin{array}{c} O_{r} \end{array}\right]}}+\underset{I_{2}}{\underset{︸}{\left[\begin{array}{cccc} 0 & -1 & 0 & 0\\ 1 & 0 & 0 & 0\\ 0 & 0 & 0 & 1\\ 0 & 0 & -1 & 0 \end{array}\right]}}⊗\underset{A_{2}}{\underset{︸}{\left[\begin{array}{c} O_{i} \end{array}\right]}}+\underset{I_{3}}{\underset{︸}{\left[\begin{array}{cccc} 0 & 0 & -1 & 0\\ 0 & 0 & 0 & -1\\ 1 & 0 & 0 & 0\\ 0 & 1 & 0 & 0 \end{array}\right]}}⊗\underset{A_{3}}{\underset{︸}{\left[\begin{array}{c} O_{j} \end{array}\right]}}\;+\underset{I_{4}}{\underset{︸}{\left[\begin{array}{cccc} 0 & 0 & 0 & -1\\ 0 & 0 & 1 & 0\\ 0 & -1 & 0 & 0\\ 1 & 0 & 0 & 0 \end{array}\right]}}⊗\underset{A_{4}}{\underset{︸}{\left[\begin{array}{c} O_{k} \end{array}\right]}}=\left[\begin{array}{cccc} O_{r} & -O_{i} & -O_{j} & -O_{k}\\ O_{i} & O_{r} & O_{k} & -O_{j}\\ O_{j} & -O_{k} & O_{r} & O_{i}\\ O_{k} & O_{j} & -O_{i} & O_{r} \end{array}\right]$$

The above equation expresses the Hamiltonian product for 4D, which preserves all PHM layer properties. When N=1, the PHM layer performs multiplication in real space.

##### Complexity Analysis of Generalized Hypercomplex CNNs

Similar to QCNN, the *N*-dimensional HCNNs (VCNNs or PHM layer) share weights across *N* input channels, so the weight-sharing ratio is 1/N for the VCNNs and PHM layer. By comparing with the standard CNN analyzed in Equation (Equation 2), the cost of VCNNs and PHM layer is,(17)$${\mathrm{Cost}}_{{\mathrm{HCNN}}_{4}}\propto \left(\frac{1}{N}\cdot h^{2}\cdot d_{in}\cdot d_{out}\cdot k^{2}\right).$$

The complexity of VCNNs is $O(\frac{1}{N}h^{2}d_{in}d_{out}k^{2})$, which is still expensive like QCNNs.

### 3.3. Modern Real-Valued CNNs

This research modifies real-valued CNNs for quaternion-based calculations. Real-valued CNNs use unit activations and weights that are real. The popularity of CNNs surged after AlexNet’s success and continued with advancements like VGG, NIN, ResNet, and DenseNet, as well as MobileNets and SqueezeNets [19] for mobile tasks.

#### 3.3.1. Residual Networks

To create very deep CNNs, convolutional layers are stacked using a residual architecture, forming residual networks (ResNets). ResNets tackle degradation issues, like vanishing gradients, that arise in deep networks. The main component is the residual block, which comes in two types: basic and bottleneck blocks. The basic block consists of two convolution layers with a kernel size of k×k (with k=3). An identity shortcut connection counters vanishing gradients. The other block in Figure 2 (right) is a bottleneck block, which consists of three convolution layers, 1×1, k×k, and 1×1, where the 1×1 layers are responsible for changing the channel count, and the 3×3 layer extracts features.

ResNets are utilized in various applications, including vision tasks, generative tasks, video processing, object detection, image segmentation, and speech enhancement [37].

##### Complexity Analysis of Modern Real-Valued CNNs

The computational cost of the residual basic block with 2 CNN layers is defined as ${\mathrm{Cost}}_{\mathrm{Basic}}\propto 2\cdot \mathrm{Cost}\;\mathrm{of}\;\mathrm{Standard}\;\mathrm{CNN}=2\cdot h^{2}\cdot d_{in}\cdot d_{out}\cdot k^{2}$. The complexity of the residual basic block is $O(2h^{2}d_{in}d_{out}k^{2})$. In terms of cost, the basic residual block has two times the complexity of the standard CNN. The cost of the pointwise 1×1 2D CNN layer is(18)$$\begin{array}{c} {\mathrm{Cost}}_{1x1\mathrm{Conv}2D}\propto h\cdot w\cdot d_{in}\cdot d_{out}=h^{2}\cdot d_{in}\cdot d_{out} \end{array}$$

The complexity of the pointwise 2D CNN layer is $O(h^{2}d_{in}d_{out})$. The cost of the residual bottleneck block is,(19)$$\begin{array}{cc} {\mathrm{Cost}}_{\mathrm{Bottleneck}} & \propto \mathrm{Cost}\;\mathrm{of}\;\mathrm{Standard}\;\mathrm{CNN}+2\cdot \mathrm{Cost}\;\mathrm{of}\;\mathrm{Pointwise}\;\mathrm{CNN}\\ \; & =h^{2}\cdot d_{in}\cdot d_{out}\cdot k^{2}+2\cdot h^{2}\cdot d_{in}\cdot d_{out}=h^{2}\cdot d_{in}\cdot d_{out}\cdot \left(k^{2}+2\right) \end{array}$$

The complexity of the residual bottleneck block is $O(2h^{2}d_{in}d_{out}+h^{2}d_{in}d_{out}k^{2})$. The cost comparison between the cost of the bottleneck residual block and the standard CNN is,(20)$$\begin{array}{cc} {\mathrm{Cost}}_{\mathrm{Ratio}} & \propto \frac{\mathrm{Cost}\;\mathrm{of}\;\mathrm{residual}\;\mathrm{bottleneck}\;\mathrm{block}}{\mathrm{Cost}\;\mathrm{of}\;\mathrm{standard}\;\mathrm{CNN}}=\frac{h^{2}\cdot d_{in}\cdot d_{out}\cdot \left(k^{2}+2\right)}{h^{2}\cdot d_{in}\cdot d_{out}\cdot k^{2}}\\ & =\frac{k^{2}+2}{k^{2}}=1+\frac{2}{k^{2}} \end{array}$$

The cost reduction in the bottleneck block (Equation (Equation 20)) is less than that of the basic block relative to the standard CNN.

#### 3.3.2. Wide Residual Networks

Wide ResNets (WRNs) [1] address performance issues by using shallow but wide architectures (shown in Figure 3b), which improve performance by widening residual blocks instead of increasing depth. Three methods to enhance the representational power of residual blocks are adding more convolutional layers per block, widening layers by increasing feature maps, and increasing filter sizes in convolutional layers. They introduced a deepening factor *l* (number of convolution layers) and a widening factor α (channel count multiplier). The notation WRN-n-α indicates a network with *n* layers and a widening factor of α. For instance, WRN-40-2 has 40 layers and twice the feature maps of the original. We compare the performance of our proposed 26-layer model with that of the WRN-28-10 network across different widening factors [1].

WRNs have been used for image classification, segmentation, super-resolution, speaker recognition, American sign language recognition, and speech recognition [37].

##### Complexity Analysis of Wide Residual Networks

The operational cost for a 2D CNN layer in WRNs is, ${\mathrm{Cost}}_{\mathrm{Conv}2D}\left(\mathrm{WRNs}\right)\propto h\cdot w\cdot d_{in}\cdot \alpha d_{out}\cdot k\cdot k,$ where α is the widening factor. Zagoruyko and Komodakis [1] experimented with the widening factor α of a block. However, it is more computationally effective to widen layers than have thousands of small kernels, as a GPU is more efficient in parallel computations on large tensors, so we are interested in an optimal *l* to α ratio [1]. We analyze the widening factor α but do not analyze this optimal ratio.

#### 3.3.3. MobileNet Architectures

For mobile and embedded vision applications, Howard et al. [21] introduced MobileNet models that use parameter-efficient depthwise separable convolutions. This method separates standard 2D convolution into depthwise and 1×1 (pointwise) convolutions. The depthwise convolution applies distinct 2D kernels to each input channel, while the pointwise convolution combines the output feature maps. This approach allows for effective feature extraction and representation, which is defined as(21)$$C_{i,j,n}=\sum_{\left(a,b\right)\in N_{k\times k(i,j)}}W_{a,b,n}X_{i+a-1,j+b-1,n},$$
where the *n*-th channel of trainable weights *W* is convolved with the *n*-th channel of input *X* to produce the *n*-th channel of the output feature map *C*. Howard et al. [21] also applied width and resolution multipliers to make the model smaller and faster, thereby reducing model complexity and performance.

MobileNet architectures have been applied to defect detection, object detection, lung cancer detection, face mask detection, facial expression recognition, image classification, crop disease identification, and breast mammogram classification [37].

##### Complexity Analysis of MobileNet Architectures

MobileNet employs depthwise separable convolution, where the number of input channels $d_{in}=1$, significantly reducing the cost of convolution by applying a single channel kernel to each input channel. The computational cost is defined like in Equation (Equation 2) as(22)$${\mathrm{Cost}}_{\mathrm{DSConv}2D}\propto h^{2}\cdot d_{out}\cdot k^{2}.$$

The complexity of a standard spatial convolution operation is $O(h^{2}d_{out}k^{2})$, which reduces costs significantly [19,21]. As the cost of MobileNets, which use the combination of depthwise and pointwise convolutions, the computational cost of MobileNets is,(23)$${\mathrm{Cost}}_{\mathrm{MobileNet}}\propto h^{2}\cdot d_{out}\cdot k^{2}+h^{2}\cdot d_{in}\cdot d_{out}.$$

It significantly reduces costs compared to the standard CNN cost in Equation (Equation 2). The cost reduction ratio by MobileNets is,(24)$${\mathrm{Cost}}_{\mathrm{Ratio}}\propto \frac{\mathrm{Cost}\;\mathrm{of}\;\mathrm{MobileNets}}{\mathrm{Cost}\;\mathrm{of}\;\mathrm{Standard}\;\mathrm{CNN}}=\frac{h^{2}\cdot d_{out}\cdot k^{2}+h^{2}\cdot d_{in}\cdot d_{out}}{h^{2}\cdot d_{in}\cdot d_{out}\cdot k^{2}}=\frac{1}{d_{in}}+\frac{1}{k^{2}}.$$

MobileNets require eight to nine times less computation than standard CNNs. They reduce the number of channels in α times and the resolution in ρ times, so the computation cost in Equation (Equation 23) can be written as [21],(25)$${\mathrm{Cost}}_{\mathrm{MobileNet}}\propto \rho h^{2}\cdot \alpha d_{out}\cdot k^{2}+\rho h^{2}\cdot \alpha d_{in}\cdot \alpha d_{out}.$$

It reduces costs more than nine times compared to standard CNNs.

#### 3.3.4. SqueezeNet Architectures

SqueezeNet architectures reduce the number of input channels in CNNs, resulting in fewer trainable parameters, less cross-server communication for distributed training, lower bandwidth requirements, and easier deployment on FPGAs with limited memory. Iandola et al. [38] proposed SqueezeNet (SNet) by squeezing the input channels to minimize the number of filters. The computational cost of the SNet convolution layer is ${\mathrm{Cost}}_{\mathrm{Conv}2D(\mathrm{SNet})}\propto h\cdot w\cdot ds_{in}\cdot d_{out}\cdot k\cdot k,$ where $ds_{in}$ is the squeezed input channels. The 1×1 convolution filter has 9× fewer parameters than the 3×3 filter. The Fire module reduces parameters by using 1×1 CNNs and limiting the number of expanded channels, allowing SqueezeNet to maintain high accuracy with a smaller model size. This design offers flexibility in balancing computational efficiency and representational power and integrates easily into larger architectures. Gholami et al. [12] reduce costs by using separable Conv2d (3×1 and 1×3) in the SqueezeNext architecture. We adapted this concept to develop parameter-efficient SqueezeNext-based network architectures.

##### Complexity Analysis of SqueezeNet Architectures

It reduces the cost once again compared to SqueezeNet, defined as(26)$${\mathrm{Cost}}_{\mathrm{SqNext}}\propto h\cdot w\cdot d_{in}\cdot d_{out}\cdot k+h\cdot w\cdot d_{in}\cdot d_{out}\cdot k=2\cdot h\cdot w\cdot d_{in}\cdot d_{out}\cdot k.$$

The SqueezeNext block is depicted in Figure 4c. And the cost reduction ratio of the SqueezeNext block compared to the standard CNN is(27)$${\mathrm{Cost}}_{\mathrm{Ratio}}\propto \frac{2\cdot h\cdot w\cdot d_{in}\cdot d_{out}\cdot k}{h\cdot w\cdot d_{in}\cdot d_{out}\cdot k\cdot k}=\frac{2}{k}.$$

#### 3.3.5. Recursive Architectures

Single Image Super-Resolution (SISR) enhances a low-resolution image into a high-resolution image. It is particularly useful in fields like satellite and medical imaging, where capturing high-frequency details is crucial. SISR outperforms reconstruction-based methods and other traditional techniques like span reduction and image upscaling [26].

##### Related Work on Recursive Networks

Deep learning models, particularly CNNs, have been applied to image super-resolution (SR) challenges. Notable contributions include VDSR (depicted in Figure 5b) and DRCN (depicted in Figure 5c), which use 20 convolutional layers and introduce techniques such as residual learning and adjustable gradient clipping to address gradient issues and manage parameter count, respectively [24,25].

However, very deep networks may be less suited to mobile systems due to their high parameter count and storage requirements. To combat this, the DRRN was developed (depicted in Figure 5d), offering improved performance with significantly fewer parameters—2× fewer than VDSR and 6× fewer than DRCN [26]. DRRN incorporates global and local residual learning as well as recursive learning of residual units to ensure a more compact model.

##### Background on Deep Recursive Residual Networks

Figure 5 presents different residual units with identity connections. Besides the “post-activation” structure, He et al. [3] propose a “preactivation” structure that performs the activation before the weight layers. The residual unit with a pre-activation structure is defined as [26]:$\;H^{u}=F\left(H^{u-1},W^{u}\right)+H^{u-1},$ where *u*=1, 2,…, *U* (*U* are the numbers of residual units in a residual block (RB)). $H^{u-1}$ and $H^{u}$ are the input and output of the *u*-th residual unit, and F denotes the residual function. The weight $W^{u}$ is shared among the residual units within a recursive block. Figure 6 shows the structure of RBs for different residual units. Here, *U* and *B* denote the numbers of residual units and residual blocks.

DRRNs with different depths is the number of convolutional layers for different B and U. The depth of DRRN d is calculated as d=(1+2×U)×B+1.

##### Complexity Analysis of Deep Recursive Residual Networks

The computational cost of the deep recursive residual network is defined as ${\mathrm{Cost}}_{\mathrm{DRRN}}\propto U\cdot B\cdot h^{2}\cdot d_{in}\cdot d_{out}\cdot k^{2}.$ The complexity of a DRRN network is $O(UBh^{2}d_{in}d_{out}k^{2})$.

## 4. Our Proposed Parameter-Efficient Architectures

Here, we introduce and analyze new methods to improve parameter efficiency in HCNNs, attention-based models, and real-valued CNNs while maintaining and sometimes improving high performance for image classification tasks.

### 4.1. Hypercomplex Based Models

As hypercomplex CNNs share weights across both channels and space, they help reduce the cost over standard CNN architectures via weight sharing across channels. The next subsections present modifications to the previous hypercomplex CNNs discussed in Section 3.2 to further improve their parameter efficiency.

#### 4.1.1. Full Hypercomplex Networks

Previous hypercomplex CNNs converted the stem and body to hypercomplex layers. By full hypercomplex, we mean also converting the real-valued backend to hypercomplex layers, which has not been done before. Although several studies have explored HCNNs (cross-channel weight sharing) [2,4,7,8,9], the cross-channel weight sharing was not used in the fully connected backend of these earlier HCNNs, so this model is novel.

Section 3.2.1 and Vectormap CNNs constructed a network frontend (stem) and intermediate stages using QCNNs and VCNNs, respectively. As they share weights across *N* channels where N=4 for QCNNs and i=1…N for VCNNs, these HCNNs reduced the parameter counts by a factor of $\frac{N-1}{N}$ [4,9]. This new hypercomplex backend gives a modest reduction in parameter count over the original QCNNs and VCNNs. The reduction size is in the thousands only [5].

The hypercomplex backend is a PHM layer (see Section 3.2.2) which replaces the standard fully connected backend. Figure 7 shows the full hypercomplex model with the PHM-based FC layer (QPHM). At the end of Stage 4 in Figure 7 (top), the output feature maps are flattened, which become the input to a fully connected layer. Depending on the dimension of the front end networks, we use a PHM layer having four dimensions with the quaternion network (QPHM) or a PHM layer having five dimensions with the three-dimensional vectormap network (introduced VPHM). Different dimensional PHM layers are needed because the output classes of any dataset must be divisible by the dimensions in the PHM operation.

Performance results of QPHMs and VPHMs are given in Section 5.1.

##### Complexity Analysis

Section Parameterized Hypercomplex Multiplication explains the linear transformation of the input $x\in R^{d_{in}}$ into an output $y\in R^{d_{out}}$. Also, this transformation uses the weight $W\in R^{d_{out}\times d_{in}}$ and bias $b\in R^{d_{out}}$, where $d_{in}$ and $d_{out}$ are input (number of units in the input layer) and output (number of units in the output layer) dimensions. Hence, the complexity of this linear transformation is $O\left(d_{in}d_{out}+d_{out}\right)=O\left(d_{in}d_{out}\right)$. However, Section Parameterized Hypercomplex Multiplication defines hypercomplex transformation using the PHM-based FC layer. $H\in R^{d_{out}\times d_{in}}$ represents the PHM layer (defined in Equation (Equation 14)) that is the sum of Kronecker product of parameter matrices $I_{i}\in R^{N\times N}$ and $A_{i}\in R^{d_{out}/N\times d_{in}/N}$, where i=1…N and *N* is the number of dimensions (N=4). Equation (Equation 16) explains the reuse of the parameter matrics $A_{i}$ and $I_{i}$. For *N*-dimensional hypercomplex PHM layer, the number of reusing parameters equals *N*. Thus, for the same input and output sizes, the parameter size of a PHM layer reduces the linear transformation 1/N times. *H* dominates parameterization and the complexity of the PHM is $O\left(d_{in}d_{out}/N+N^{3}\right)=O\left(d_{in}d_{out}/N\right)$, where $d_{in}d_{out}\geq N^{4}$ (in this experiment $d_{in}=2048,d_{out}=100,N=2,4,8$).

#### 4.1.2. Separable Hypercomplex CNNs

In another approach to reduce costs, this section introduces novel separable HCNNs. Specifically, we propose another hypercomplex-based network to reduce parameters more than the models of Section 4.1.1 by introducing separable hypercomplex networks (SHNNs) [22]. These SHNNs (1) handle multidimensional inputs; (2) apply weight sharing across input channels; (3) capture cross-channel correlations; (4) reduce computational costs; and (5) increase validation accuracy performance for image classification datasets.

Figure 8a shows a separable hypercomplex block. The QCNN layers in Figure 7 are replaced by the SHNN block in Figure 8a to construct SHNNs. This separable concept has also been used for real-valued networks [12,39] as explained in Section 3.3.3. They decompose the quaternion convolutional operation into two consecutive separable vectormap convolutional operations, splitting 2D spatial filters with 3×3 into two separable filters with 3×1 and 1×3. These separable filters are applied to the height and width axis of inputs. To apply the separable concept, we replace the 3×3 spatial quaternion convolution with two separable VCNN layers sequentially to the height axis (three channels 3×1 VCNN layer) and width axis (3 channels 1×3 VCNN layer). This decomposition helps to reduce cost from $k^{2}$ to 2k.

The two 1×1 quaternion convolutional layers remain unchanged like the original QCNNs [9], which are responsible for reducing and increasing the number of channels. These help to reduce $\frac{N\;-\;1}{N}$ parameters. This forms our proposed SHNN bottleneck block seen in Figure 8a, stacked multiple times to construct separable hypercomplex ResNets (SHNNs). Each quaternion and vectormap convolution accepts four and three input channels and produces four and three output channels, respectively. Hence, the required input and output channels must be divided by three and four [4,18,22].

Performance results on SHNNs are given in Section 5.1.

##### Complexity Analysis

For vision tasks, HCNNs (including QCNNs and VCNNs) take $O(\frac{1}{N}h^{2}d_{in}d_{out}k^{2})$ resources for a 2D image with HCNN dimension *N*, the input channel count $d_{in}$, the output channel count $d_{out}$, kernel k×k, height *h*, width *w*, and h=w.

Separable hypercomplex models work on one dimension of a feature map at a time, but the input images are two-dimensional. A square 2D input is split into height and width axes for a two-dimensional feature map. As the 3-channel VCNN operation is first applied along the height axis of the input image and then the width axis, the computational cost is reduced to $O(\frac{2}{N}hd_{in}d_{out}k)$ from the HCNN cost of $O(\frac{1}{N}h^{2}d_{in}d_{out}k^{2})$. This cost reduction reveals our parameter-efficient deep learning networks in deep learning hypercomplex space.

### 4.2. Attention-Based Models

Attention networks have been used for visual classification, but they tend to be more computationally expensive than CNNs. Wang et al. [11] introduced axial attention networks to reduce these costs. Their proposed mechanism, called Axial-ResNet, factorizes a 2D self-attention operation into consecutive 1D self-attention operations, similar to the axial CNNs described in the introduction. Ref. [18] modified this design by introducing a HCNN layer that discovers and represents cross-channel correlations in its input. These correlations lead to feature maps with improved representational properties. In this new model [18], a hypercomplex layer is added to the original Axial-ResNet block [11].

Figure 8b shows the original Axial-ResNet design, and Figure 8c shows our modification with a 1×1 quaternion convolution layer added before the axial attention module. This additional layer helps generate useful interlinked input representations. These better representational feature maps improve the performance of the axial-attention module. Although this new block improves performance, the additional layer increases the layer count for each block of this kind. It was hypothesized that there may be some redundant computations within this new block.

To address this, ref. [18] introduced another block type by removing the additional 1×1 quaternion CNN module from our first proposal (Figure 8c) and replacing 1×1 convolutional down-sampling and up-sampling modules with a bank of 1×1 quaternion 2D convolution modules. Figure 8d shows the final modification, named representational axial-attention block (RepAA). We replaced the QCNN blocks in the stem and body of Figure 7 using RepAA blocks to construct a quaternion enhanced axial attention network. We chose this network (Figure 7), as it has a QCNN-based stem layer (the first layer of the network) as a quaternion-based front-end layer and a PHM-based fully connected dense layer as a quaternion (PHM layer with four dimensions)-based backend layer.

Performance results for the RepAA network are shown in Section 5.2.

#### Complexity Analysis of Attention-Based Models

As the computational cost of HCNNs is reduced to $\frac{N-1}{N}$ where *N* is the HCNN dimension, for vision tasks, the representational axial-attention networks reduce to $O(\frac{1}{N}h^{2}d_{in}d_{out}k)$ resources for a 2D image in comparison to the original axial-attention cost of $O(h^{2}d_{in}d_{out}k)$.

### 4.3. Convolution Based Models

Section 4.1 and Section 4.2 explain parameter-efficient hypercomplex inspired networks and attention networks, respectively. This section introduces more parameter-efficient CNN architectures (which are not hypercomplex) for computers and mobile devices in real-valued space.

#### 4.3.1. Computer Based Networks

Computer-based deep learning networks, also known as deep neural networks (DNNs), are complex architectures that enable machines to learn from data by mimicking the way the human brain works. They consist of multiple layers of interconnected nodes (neurons) that process and transform input data to extract features and make predictions. On the other hand, a mobile-embedded deep learning network refers to the integration and deployment of deep learning models on mobile devices and embedded systems. This approach enables real-time processing, decision-making, and inference directly on devices with limited computational resources, such as smartphones, tablets, IoT devices, and edge computing hardware. The goal is to bring the power of deep learning to devices that operate in resource-constrained environments, allowing for applications that require low latency, offline processing, and enhanced privacy. As standard 2D convolution layers consume relatively high computational costs, several modified architectures have been constructed to reduce these costs. Among them, residual bottleneck blocks use 1×1 convolutions to reduce input channel counts, so fewer channels are processed [3]. They still use standard 2D convolutions. These convolutions are also used in wide ResNets [1].

The success of CNNs like ResNets [3], wide ResNets [6], scaling wide ResNets [1], and deep recursive residual networks (DRRNs) [26] is demonstrated on image classification and image super-resolution datasets. For vision tasks, the input image is $X\in h\times w\times d_{in}$, where *h*, *w*, and $d_{in}$ are the height, width, and input channel count of an input image. Equation (Equation 1) defines the operation of standard 2D convolution, residual basic block, and residual bottleneck block. The cost of standard 2D convolution, residual basic, and bottleneck blocks is calculated in Equations (Equation 2) and (Equation 19), respectively. The separable convolutional operation reduces computational costs in InceptionNetV3 [40] and axial-deeplab [11].

Separable convolution is a convolution operation that decomposes a standard convolution into two separate convolutions which reduces the computational complexity and increases efficiency. Axial convolution is a more recent approach that decomposes the convolution operation across individual axes (height, width, or depth) of the input data. This technique focuses on enhancing the model’s ability to capture long-range dependencies and context information along specific dimensions.

Separable convolution splits the k×k convolutional filter into two filters with sizes k×1 and 1×k. This decomposition effectively reduces the number of parameters from $h^{2}\times d_{in}\times d_{out}\times k^{2}$ to $2\times h^{2}\times d_{in}\times d_{out}\times k$. The convolution operation with spatial extent k×1 is defined as [19],(28)$$C_{i,j,n}=\sum_{\left(a,b,m\right)\in N_{k\times 1(i,j)}}W_{a,b,m,n}X_{i+a-1,j+b-1,m}$$
where *m* and *n* are the indices for input channel $d_{in}$ and for output channel $d_{out}$. Also, $N_{k}\in R^{k\times 1\times d_{in}}$ is the neighborhood of pixel (i,j) with spatial extent k×1, and $W\in R^{k\times 1\times d_{out}\times d_{in}}$ is the shared weights that are for calculating the output for all pixel positions (i,j). For spatial extent 1×k, the convolution is defined as [19](29)$$C_{O(i,j,n)}=\sum_{\left(a,b,m\right)\in N_{1\times k(i,j)}}W_{a,b,m,n}C_{i+a-1,j+b-1,m}$$
where $N_{k}\in R^{1\times k\times d_{in}}$ is the neighborhood of pixel (i,j) with extent 1×k and $W\in R^{1\times k\times d_{out}\times d_{in}}$ are the shared weights that calculate the output for all pixel positions (i,j).

The costs of these decomposed/separable convolutional operations are still high, as they use standard 2D convolutions. To reduce these costs, we introduce a novel residual 1D convolutional network (RCN) block that replaces the standard 2D convolutional layers with RCN blocks (see Figure 9) [19]. RCN blocks use two consecutive 1D depthwise separable convolution (DSC) operations with *k* filters instead of a standard 2D convolution. Figure 9 shows the RCN block with two 1D depthwise convolution layers followed by a residual connection to avoid vanishing gradients. As 1D convolution is used, the inputs h×w split into the height *h* and width *w* axes. Each 1D convolution layer applies to each input axis. The 1D DSC operation proposed is defined as(30)$$C_{O(i,n)}=\sum_{a\in N_{k(i)}}W_{a,n}X_{i+a-1,n}$$
where $N_{k}\in R^{k\times d_{in}}$ is the neighborhood of pixel *i* with extent *k* and $W\in R^{k\times d_{out}\times d_{in}}$ is the shared weights that calculate the output for all pixel positions *i*.

Here, the *n*-th channel of trainable weights is applied to the *n*-th channel of input to produce the *n*-th channel of the output feature map $C_{O}$. The cost of this 1D DSC operation is(31)$${\mathrm{Cost}}_{\mathrm{Conv}1D}\propto h\cdot d_{out}\cdot k.$$

As our RCN block has two 1D convolution layers, this block costs $2\cdot h\cdot d_{out}\cdot k$. This block replaces the 2D standard convolution layer in the residual, WRN basic, and bottleneck blocks and constructs RCN-based basic and bottleneck blocks. Also, the standard 2D convolution in deep recursive residual networks is replaced by the RCN block and introduced recursive 1D convolutional residual networks (RCRNs) shown in Figure 10. The computational cost of the original deep recursive residual network is given in Section Complexity Analysis of Deep Recursive Residual Networks where the cost depends multiplicatively on the residual units *U*, residual blocks *B*, the number of input channels $d_{in}$, the number of output channels $d_{out}$, kernel size k×k, and input feature map size $h\times w=h^{2}$ for vision tasks. The cost of the RCRNs is(32)$${\mathrm{Cost}}_{\mathrm{RCRN}}\propto 2\cdot U\cdot B\cdot h\cdot d_{out}\cdot k.$$

The complexity of RCRN is $O(UBhd_{out}k)$. Performance results of RCRNs are given in Section 5.3.

#### 4.3.2. Mobile Embedded Networks

We also introduce real-valued parameter-efficient RCNs for mobile and embedded vision tasks [19]. Compared with other mobile-based networks, the depthwise 2D spatial convolution (marked red in Figure 4a) of MobileNet is replaced by the RCN block and constructs a new RCN-based MobileNet block. Also, compared with separable convolution, we replace the red-marked area in Figure 4c of SqueezeNext with the RCN block and construct the RCN-based SqueezeNext block. These RCN-based MobileNet and SqueezeNext blocks are building blocks of the RCN-based MobileNets and SqueezeNexts (network architectures).

The performance results of these proposed networks are given in Section 5.3.

#### 4.3.3. Complexity Analysis of Mobile Embedded Convolutional Networks

This section explains how RCNs are more cost effective than the models for both standard and mobile-embedded devices. The costs of the standard convolutional operation, residual basic blocks, bottleneck blocks, WRN convolutional operation, MobileNet block, SqueezeNet, and SqueezeNext block are calculated in Equations (Equation 2), (Equation 19), (Equation 25) and (Equation 26). The computational costs of the RCN block compared with the standard 2D convolutional operation are [19],(33)$${\mathrm{Cost}}_{R}\propto \frac{\mathrm{Cost}\;\mathrm{of}\;2D\;\mathrm{Convolution}}{2\cdot \mathrm{Cost}\;\mathrm{of}\;1D\;\mathrm{Convolution}}=\frac{h\cdot w\cdot d_{in}\cdot d_{out}\cdot k^{2}}{2\cdot h\cdot d_{out}\cdot k}=\frac{w\cdot d_{in}\cdot k}{2}$$
where ${\mathrm{Cost}}_{R}$ defines the cost reduction ratio where the RCN block reduces costs $(w\cdot d_{in}\cdot k)/2$ times compared to the original standard 2D convolution. We construct the RCN-based residual basic block (replacing the two 2D CNN layers) and the RCN-based residual bottleneck block (replacing the only spatial 2D CNN layer) [19]. The RCN-based ResNet basic block reduces the costs compared with the residual basic block of(34)$${\mathrm{Cost}}_{R}\propto \frac{2\cdot \mathrm{Cost}\;\mathrm{of}\;2D\;\mathrm{Convolution}}{2\cdot \mathrm{Cost}\;\mathrm{of}\;1D\;\mathrm{Convolution}}=\frac{2\cdot h\cdot w\cdot d_{in}\cdot d_{out}\cdot k\cdot k}{2\cdot h\cdot d_{out}\cdot k}=w\cdot d_{in}\cdot k$$
where ${\mathrm{Cost}}_{R}$ defines the cost reduction ratio, where the RCN-based ResNet basic block reduces costs $\left(w\cdot d_{in}\cdot k\right)$ times than the original ResNet basic block. For the ResNet bottleneck block, the RCN-based bottleneck block reduces costs like the cost reduction in Equation (Equation 33). The 2D CNN layers in DRRNs are replaced by the RCN block and constructed RCRNs. The costs comparison between DRRN in Section Complexity Analysis of Deep Recursive Residual Networks and RCRN in Equation (Equation 32) is(35)$${\mathrm{Cost}}_{R}\propto \frac{\mathrm{Cost}\;\mathrm{of}\;\mathrm{DRRN}}{\mathrm{Cost}\;\mathrm{of}\;\mathrm{RCRN}}=\frac{U\cdot B\cdot h^{2}\cdot d_{in}\cdot d_{out}\cdot k^{2}}{2\cdot U\cdot B\cdot h\cdot d_{out}\cdot k}=\frac{h\cdot d_{in}\cdot k}{2}$$
where ${\mathrm{Cost}}_{R}$ defines the cost reduction ratio where the RCRN reduces costs $(h\cdot d_{in}\cdot k)/2$ times than the original DRRN.

We also analyze the cost effectiveness of MobileNet and SqueezeNext architectures with our proposed RCN block-based MobileNet and SqueezeNext architectures [19]. The RCN-based MobileNet block performs a reduction in the costs of(36)$${\mathrm{Cost}}_{R}\propto \frac{{\mathrm{Cost}}_{\mathrm{DWConv}2D}\;\mathrm{in}\;\mathrm{MobileNetV}1}{2\cdot \mathrm{Cost}\;\mathrm{of}\;1D\;\mathrm{Convolution}}=\frac{h\cdot w\cdot d_{out}\cdot k\cdot k}{2\cdot h\cdot d_{out}\cdot k}=\frac{w\cdot k}{2}$$
where ${\mathrm{Cost}}_{R}$ defines the cost reduction ratio where the RCN-based MobileNet reduces costs (w·k)/2 times than the original MobileNets and $d_{in}$ is 1 for the original and parameter-efficient spatial convolutions, as both networks use depthwise separable convolutions. For SqueezeNext architectures, the RCN-based SqueezeNext reduces the computational costs of(37)$$\begin{array}{cc} {\mathrm{Cost}}_{R} & \propto \frac{\mathrm{PW}1x1\mathrm{Conv}2D\;+\;2\cdot \mathrm{kx}1\mathrm{Conv}2D}{2\cdot \mathrm{Cost}\;\mathrm{of}\;1D\;\mathrm{Convolution}}\\ & =\frac{h\cdot w\cdot d_{in}\cdot d_{out}+2\cdot h\cdot w\cdot d_{in}\cdot d_{out}\cdot k}{2\cdot h\cdot d_{out}\cdot k}=\frac{w\cdot d_{in}}{2\cdot k}+w\cdot d_{in} \end{array}$$
where ${\mathrm{Cost}}_{R}$ is the cost reduction ratio of the original SqueezeNext and the RCN-based SqueezeNext blocks. The RCN-based SqueezeNext block takes $\left(w\cdot d_{in}\right)/\left(2\cdot k\right)+\left(w\cdot d_{in}\right)$ times less computing from separable convolutional operations in the original SqueezeNext block. These comparisons show that the RCN-based block is parameter efficient and cost effective in replacing 2D convolutions for vision tasks.

This section examines the cost efficiency of our proposed computer-based and mobile embedded models compared to relevant network architectures.

## 5. Performance Comparisons for Our Proposed Models

This section presents experimental results on three novel CNN types: hypercomplex CNNs (Section 5.1), axial-attention-based CNNs (Section 5.2), and real-valued CNNs in visual classification tasks (Section 5.3). First, we compare the SHNNs with existing convolution-based hypercomplex networks, such as complex CNNs, QCNNs, octonion CNNs, and VCNNs. Then, we compare the representational axial attention networks with original axial-attention networks, QCNNs, and ResNets. Finally, we compare the residual conv1d networks with other related real-valued networks, such as original ResNets, wide ResNets, mobileNets, and squeezeNexts. All comparisons are performed on several image classification tasks. Our results primarily focus on parameter counts and performance.

### 5.1. Performance of Hypercomplex-Based Models

This section presents results that are related to HCNNs and are parameter efficient in hypercomplex space. Table 1 shows the performance of different architectures along with the number of trainable parameters on CIFAR benchmarks (CIFAR10 and CIFAR100) [41], SVHN [42], and tiny ImageNet [43] datasets. This table compares 26-, 35-, and 50-layer architectures of ResNets [3], QCNN [9], VCNN [4], QPHM [5], VPHM [5], and SHNN [22]. It also compares hypercomplex networks with full HCNNs, i.e., QPHM and VPHM, which stand for QCNNs with hypercomplex backend and VCNN with hypercomplex backend. Although the full HCNNs showed better performance in hypercomplex areas, they did not significantly reduce the parameter count and other computational costs.

We compare our proposed SHNNs with full HCNNs and other HCNNs in regard to parameter count, FLOPS, latency, and validation accuracy results, shown in Table 1. This table and Figure 11a,b show that SHNNs outperform in validation accuracy with lower parameter count and FLOPs for all image classification datasets. Figure 12a,b show the accuracy and loss curves for 50 layer ResNet, QCNN, QPHM and SHNN. We also compare the performance of SHNNs with other HCNNs where the SHNNs show state-of-the-art results for CIFAR benchmarks, shown in Table 2.

For an input image with sizes height *h*, and width *w*, the HCNNs take $O\left(N^{2}\right)=O\left({\left(hw\right)}^{2}\right)=O\left(h^{2}w^{2}\right)=O\left(h^{4}\right)$ resources for an image of length *N* where *N* is the flattened pixel set N=hw, and h=w. The SHNN reduces these computational costs and is constructed in a parameter-efficient manner. To construct SHNNs, the 3×3 spatial QCNN operation is replaced by two 3-channels 3×3 VCNN layers. The 3-channel VCNN operation is first applied along the 1D input image region of length *h* and then applied along the 1D input image region of length *w*. These two 1D operations that are finally merged together reduce the cost to $O\left(h\cdot h^{2}\right)=O\left(h^{3}\right)$ from the HCNNs cost of $O(h^{4})$.

However, the SHNN latency is slightly higher than that of other HCNNs due to the transition from 4D HCNN to 3D HCNNs and 3D HCNNs to 4D HCNN [22]. The latency for VCNNs is also high.

### 5.2. Performance of Axial Attention-Based Models

This section presents experimental results for attention-based parameter-efficient architectures in visual classification tasks. The comparisons involve hypercomplex axial-attention networks [18] using ResNets [3], QCNNs [9], QPHM [5], and axial-attention networks [11], and are shown in Table 3 for the ImageNet300k dataset. Table 3 compares 26- layer (33-layer for QuatE-1 as QuatE-1 added an extra 1×1 quaternion CNN bank module), 35-layer, and 50-layer (66-layer for QuatE-1, as QuatE-1 added an extra 1×1 quaternion CNN bank module) architectures.

Although the layer count is increased for quaternion-enhanced (QuatE) axial-attention networks (26-layer becomes 33-layer networks, and 50-layer becomes 66-layers), the performance of the 35-layer networks does not improve enough to surpass the QuatE (33-layer) version. A similar situation happens for the 50-layer (66-layer for QuatE) version. The quaternion front end has more impact (the quaternion modules produce more usable interlinked representations) than the layer count [18] but uses more parameters. To address this, we use a quaternion-based dense layer (QPHM) in the backend of QuatE axial attention networks [18]. As explained in Section 4.1.1, this reduces parameter counts slightly.

The most impressive results come from our novel axial-attention ResNets (RepAA) explained in Section 4.2. We propose another quaternion-based axial attention ResNet named representational (because of improved feature map representations) axial-attention ResNets (RepAA) to reduce parameters significantly [18].

The most important comparison is the RepAA networks with other attention-based ResNets (QuatE axial-ResNet and axial-ResNet). It directly shows the effect of applying representational effects throughout the attention network. In all architectures, excluding 35 layers where our first proposed architecture (QuatE-1) perform better, the RepAA networks outperform in classification accuracy with far fewer parameters. This solves the problem encountered in QuatE-1 networks (discussed in the previous paragraph) and supports our analysis of parameter-efficient attention models, as our proposed RepAA network architectures consumes fewer parameters than the other relevant network architectures, shown in Table 3.

### 5.3. Performance of Real-Valued CNNs

This section presents results on parameter-efficient real-valued computer and mobile-embedded CNNs (explained in Section 4.3), and parameter-efficient convolutional vision transformers. We compare ResNets and RCNs of 26,35,50,101, and 152-layer residual CNN architectures. We also compare MobileNet with parameter-efficient RCMobileNet, and SqueezeNext (23 layers with widening factors 1 and 2) with parameter-efficient RCSqueezeNext architectures.

We start by showing comparisons between ResNets and RCNs in Table 4. The experiments use the CIFAR benchmarks [41], Street View House Number (SVHN) [42], and Tiny ImageNet [43] image classification datasets. We compare a range of shallow and deep architectures consisting of 26,35,50,101, and 152 layers. Our comparison for this part of the paper is in terms of parameter count, FLOPS count, latency, and validation accuracy on the four datasets.

Table 4 describes that the 26,35,50,101, and 152-layer RCN architectures reduce by 77%, 76.9%, 76.7%, 76.6%, and 76.5% trainable parameters, respectively, and 15 to 36 percent fewer FLOPS in comparison to real-valued convolutional ResNets. We propose parameter-efficient RCNs that improve the validation performance significantly for all network architectures mentioned in Table 4 than the original ResNets for all datasets [19]. We demonstrate the layer-wise performance improvement as “the deeper, the better” in classification. The parameter-efficient RCN-based ResNets consume 39.5%, 38%, 35%, 24%, and 23.8% fewer latency than the original ResNets for 26,35,50,101, and 152-layer architectures, respectively. These costs state the parameter-efficient computer-based networks.

We present results examining the widening factors for residual networks [19] in Table 5. For a fair comparison with WRNs [1], we use 26-layer RCNs with widening factors 2,4,6,8, and 10 to 28-layer WRNs. WRCNs outperform the original WRNs for all widening factors shown in Table 5. Shahadat and Maida [19] show that our 26-layer proposed WRCNs consume 86% fewer parameters than the original WRN [1] and demonstrate “the wider, the better”.

We also analyze mobile-embedded architectures with our parameter-efficient RCN blocks (see Figure 9) [19]. We apply our RCN concept in MobileNet and SqueezeNext architectures to propose RCMobileNet and RCSqueezeNext architectures. As seen in Table 4, the RCMobileNet achieves more than validation accuracy on CIFAR10 with 75% fewer trainable parameters and almost 67% fewer FLOPs than the original MobileNet architecture. Our RCMobileNet has a similar latency to the original MobileNet. Table 4 compares the performance for 23-layer SqueezeNext architectures with widening factors 1 and 2. Our RCSqueezeNexts consume 34%, 24%, and 33% fewer parameters, FLOPs, and latency, respectively, and show better validation performance than the original SqueezeNexts [19].

We also apply our proposed RCN modules to the CMT and DRRN architectures and proposed parameter-efficient RCMT and RCRN for image classification and image super-resolution datasets. The performance comparisons of CMT architectures are in Table 4 for CIFAR benchmarks and SVHN datasets. We compare the Peak Signal-to-Noise Ratio (PSNR) results of our parameter-efficient proposed RCRN with different networks on image super-resolution dataset [19]. We compare DRRN and RCRN, as they directly indicate the effectiveness of parameter-efficient RCN blocks. For fair comparison, we construct RCRN19 ($B=1,U_{RCN}=9$) and RCRN125 ($B=1,U_{RCN}=25$) to compare with the original DRRN19 (B=1,U=9) and DRRN125 (B=1,U=25), respectively [19]. Our comparisons are shown in Table 6 on the Set5 dataset and for all scaling factors. Also, our parameter-efficient proposed RCRN19 takes 18182 parameters compared to the 297,216 parameters of DRRN19.

### 5.4. Basis for Selecting the 1D Convolution Kernel Size in RCNs

Residual one-dimensional convolutional networks (RCNs) replace the spatial 2D convolutional layer with two sequential 1D depthwise separable convolution (DSC) operations, each using a kernel of size. Because the receptive field of each 1D convolution is controlled entirely by *k*, the choice of the kernel size is central to the representational capacity and efficiency of the RCN block, defined in Equation (Equation 30). The effective receptive field of each RCN block is therefore determined by the depthwise kernel size *k* applied independently along the height and width axes.

#### 5.4.1. Rationale Behind Kernel Size Selection

The kernel size *k* is selected to preserve the spatial modeling capability of standard 2D convolution while significantly reducing computational cost. A conventional 2D convolution with a kernel of size k×k incurs a computational cost, shown in Equation (Equation 2), whereas a single 1D depthwise convolution in an RCN has linear complexity, shown in Equation (Equation 31). Because the RCN block applies two such 1D operations, its total cost becomes ${\mathrm{Cost}}_{RCN}\propto 2h\;d_{out}\;k,$ resulting in a substantial reduction in complexity relative to standard 2D convolution. Consequently, *k* is chosen to ensure sufficient spatial coverage, analogous to 3 × 3 or 5 × 5 filters in 2D convolution, while maintaining the linear computational growth that characterizes the 1D formulation.

#### 5.4.2. Parameter and Computational Cost Implications

The cost reduction ratio between a 2D convolution and the RCN block is expressed as $\mathrm{Reduction}\;\mathrm{Ratio}=\frac{hwd_{in}d_{out}k^{2}}{2hd_{out}k}=\frac{w\;d_{in}\;k}{2},$ demonstrating that RCNs reduce computation by a factor proportional to $\frac{w\cdot d_{in}\cdot k}{2}$. Because RCN complexity grows linearly with *k*, in contrast to the quadratic $k^{2}$ growth of 2D convolution, larger kernel sizes can be used without incurring prohibitive parameter or FLOP increases. This property is particularly advantageous for mobile and embedded applications, where computational budgets are limited.

#### 5.4.3. Summary of Selecting the 1D Convolution Kernel Size in RCNs

The selection of the 1D convolution kernel size in RCNs is guided by the need to preserve spatial representational power while exploiting the linear-cost benefits of 1D depthwise separable convolution. Larger kernel sizes improve accuracy by expanding the receptive field, while RCN linear scaling ensures that computational costs remain manageable. This design principle enables RCN-based architectures to deliver strong classification performance with significantly reduced parameter counts, FLOPs, and latency compared with standard 2D convolutional networks.

### 5.5. Ablation Study on 1D Convolution Kernel Size in RCN-Based Architectures

This section presents an ablation analysis that evaluates the impact of the size of the 1D convolution kernel *k* on classification accuracy and parameter efficiency in two architectures: RCMobileNet and RCSqueezeNext. Since RCNs replace each spatial 2D convolution with two sequential 1D depthwise separable convolutions, the kernel size directly determines the effective 1D receptive field and plays a fundamental role in controlling both accuracy and model complexity.

The results indicate that increasing *k* consistently improves accuracy on CIFAR-10 and CIFAR-100 up to a saturation point. The linear dependence of parameter count on *k* allows larger kernels to be used without significantly increasing model size, in contrast to the quadratic $k^{2}$ scaling found in 2D convolutions.

#### 5.5.1. Kernel-Size Effect in RCMobileNet

Table 7 evaluates the effect of varying *k* in RCMobileNet. Moderate kernel enlargement (up to k=7) improves accuracy with minimal increases in parameter count, demonstrating that the RCN substitution for MobileNet’s depthwise convolution remains highly efficient.

#### 5.5.2. Kernel-Size Effect in RCSqueezeNext

Table 8 presents the ablation results for RCSqueezeNext. As with other architectures, kernel sizes up to k=7 produce consistent accuracy gains, while larger kernel sizes yield diminishing returns.

Overall, across all two architectures, kernel sizes in the range of k=5 to k=9 offer the best balance between accuracy and efficiency for CIFAR-scale data.

## 6. Summary and Conclusions

This paper analyzed the parameter efficiency of many widely used models, including hypercomplex CNNs, attention-based CNNs, and real-valued computer and mobile-embedded CNNs on vision classification tasks. This information is essential for those interested in implementations of parameter-efficient models.

The cross-channel weight-sharing concept used in hypercomplex filters, along with the Hamiltonian product, helps to implement parameter-efficient hypercomplex convolutional networks. Also, this weight-sharing property of HCNNs allows building models as cost-effective architectures, which is a perfect fit for three- or four-dimensional input features. Although traditional HCNNs form a good foundation for parameter-efficient implementations [4,9,15], our efficiency can be improved as all models use the real-valued backend in their architectures. We applied hypercomplex properties throughout the network, including the backend dense layer, to construct a full hypercomplex-based model. They applied the PHM-based dense layer in the backend of hypercomplex networks, which helped to imply hypercomplex concepts throughout the network. This novel design improved classification accuracy, and reduced parameter counts, and FLOPs more than the other related networks. The results [5] support our analysis that the PHM operation in the densely connected backend implements parameter-efficient HCNNs, but it did not reduce parameters and FLOPs significantly.

To further reduce parameter counts, we proposed separable HCNNs by splitting the 2D convolutional (3×3) filter into two separable filters with size of 3×1 and 1×3. To apply separable filters, we replaced the quaternion operation in the quaternion CNN block using two separable vectormap convolutional networks and formed the SHNN block. This separable hypercomplex bottleneck block improves classification performance on the CIFAR benchmarks, SVHN, and Tiny ImageNet datasets compared to the baseline models. These SHNNs also show state-of-the-art performance in hypercomplex areas and reduce parameter counts and FLOPS significantly. But they have longer latency than real-valued and hypercomplex-valued CNNs. The reason behind this higher latency is the model performs convolution twice and takes transition time from 2D convolution to two consecutive 1D convolutions. The parameter and FLOP reduction support our analysis that the SHNNs are parameter-efficient HCNNs.

Although axial-attention networks are parameter-efficient attention-based networks, they still use convolution-based spatial weight sharing. We applied cross-channel weight sharing along with spatial weight sharing to the attention-based models. They applied representationally coherent modules generated by HCNNs in the stem, the bottleneck blocks, and the fully connected backend [18]. These results are important because the improvement was observed when the hypercomplex network was applied for the axial attention block. This suggests that this technique may be useful in reducing parameters and making parameter-efficient attention-based networks.

We also reviewed real-valued models like residual networks, vision transformers, and mobile-supported MobileNet and SqueezeNext architectures. Our comparison is limited to classification-based vision models. We compared the RCN block, constructed with two sequential 1D DSCs and residual connections, with original ResNets for different layer architectures. These modifications helped to reduce by around 80% the number of trainable parameters, around 30% FLOPs, and around 35% latency, as well as improving validation performance on image classification tasks. This cost reduction supports our hypothesis that the RCN is a parameter-efficient architecture.

We also checked our proposed RCN block for ResNets, WRNs, MobileNet, and SqueezeNext architectures, which showed that the RCNs-based ResNets, WRCNs, MobileNet, and SqueezeNext take fewer parameters, FLOPs, and latency and obtain better validation accuracy on different image classification tasks. We also applied our proposed method to DRRNs that improve PSNR results on image super-resolution datasets and reduce around 94% trainable parameters compared to the other CNN-based super-resolution models. These suggest that the RCN block helps construct parameter-efficient real-valued computer, mobile, and vision transformer embedded networks for vision tasks.

Future research can extend the proposed parameter-efficient CNN architectures in several meaningful directions. First, while this study focused on image classification tasks, applying SHNN, RepAA, and RCN models to other domains such as object detection, semantic segmentation, or even few-shot learning could test their adaptability and generality. Second, integrating additional compression techniques such as quantization, pruning, and knowledge distillation with the current architectures may further reduce model size and computational requirements. Evaluating the performance of these networks on real-world edge devices like mobile GPUs or microcontrollers would validate their practicality, particularly given the RCN design for embedded systems. It would also be valuable to test the cross-domain robustness of these models by applying them to fields like medical imaging or satellite image analysis. Finally, future work could explore the integration of lightweight and hypercomplex CNN components into transformer-based or multimodal vision architectures, opening paths for more efficient vision–language models. These directions build on the core principle of structural efficiency and weight sharing demonstrated in SHNN, RepAA, and RCN, aiming to broaden their impact across both tasks and platforms.

## Figures and Tables

**Figure 1 sensors-25-07663-f001:**
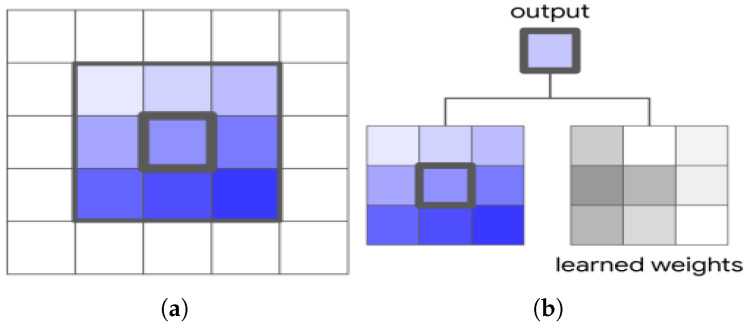
Convolutional operation. (**a**) A local window around i=3,j=3 with spatial extent k=3, and (**b**) 3×3 CNN operation between input $X_{a,b}$ and learnable weights $W\in R^{3\times 3\times d_{out}\times d_{in}}$ [19].

**Figure 2 sensors-25-07663-f002:**
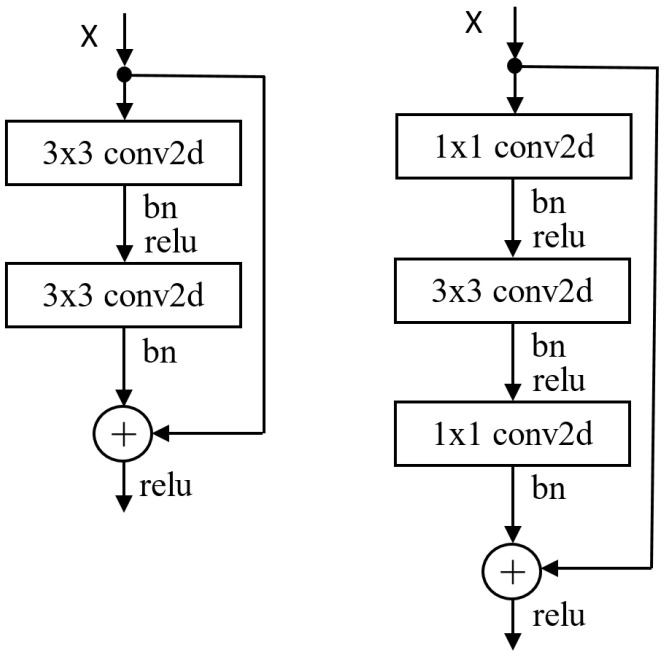
Residual block architectures. “bn” stands for batch normalization. (**Left**) ResNet basic and (**Right**) bottleneck blocks are shown.

**Figure 3 sensors-25-07663-f003:**
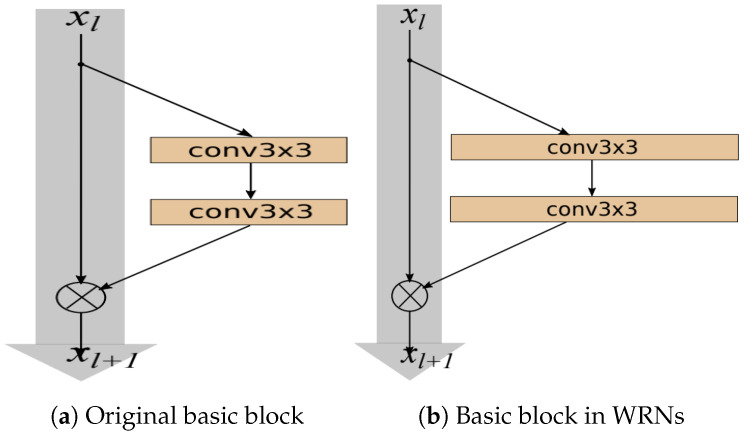
Illustration of an original basic block in a ResNet [3] (**Left**), and a basic block in WRNs [1] (**Right**). The CNN layers in the right block produce wider channels than the left basic block [1].

**Figure 4 sensors-25-07663-f004:**
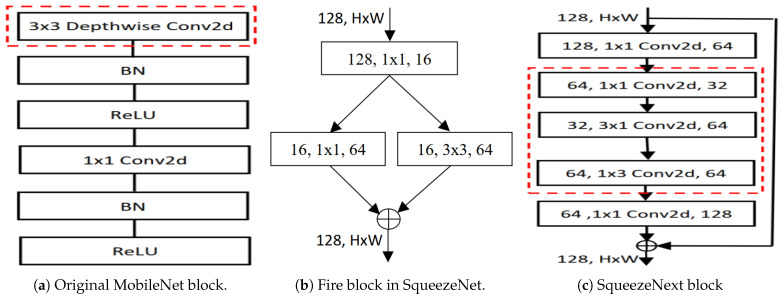
Illustration of an (**a**) Original MobileNet block found in [21]. “bn”, “ReLU”, and “Conv2d” stand for batch normalization, rectified linear unit, and 2D CNN, respectively, (**b**) SqueezeNet block [38], and (**c**) SqueezeNext block [12].

**Figure 5 sensors-25-07663-f005:**
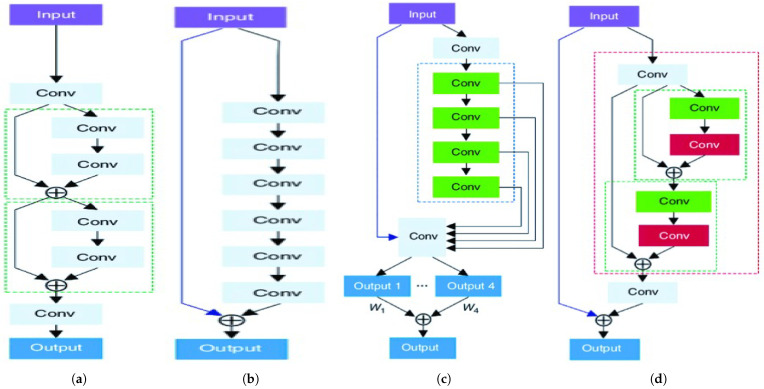
Simplified structures of (**a**) Residual network (ResNet) [3]. The green dashed box means a residual unit. (**b**) VDSR [25]. The purple line refers to global identity mapping. (**c**) DRCN [24]. The blue dashed box refers to a recursive layer, among which the convolutional layers (with light green color) share the same weights. (**d**) DRRN [26]. The red dashed box refers to a recursive block of two residual units. In all four cases, the outputs with light blue color are supervised, and ⨁ is element-wise addition. These figures are collected from [26].

**Figure 6 sensors-25-07663-f006:**
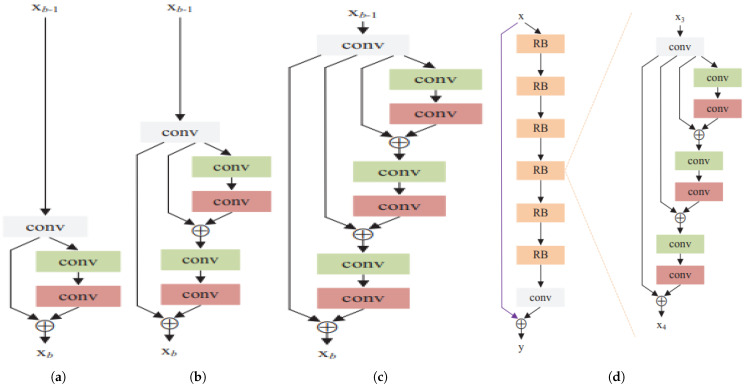
Structures of deep recursive residual blocks. U means the number of residual units in the recursive block. These blocks are collected from [26]. (**a**) U = 1; (**b**) U = 2; (**c**) U = 3; (**d**) DRRN for 6 recursive blocks and 3 residual units.

**Figure 7 sensors-25-07663-f007:**
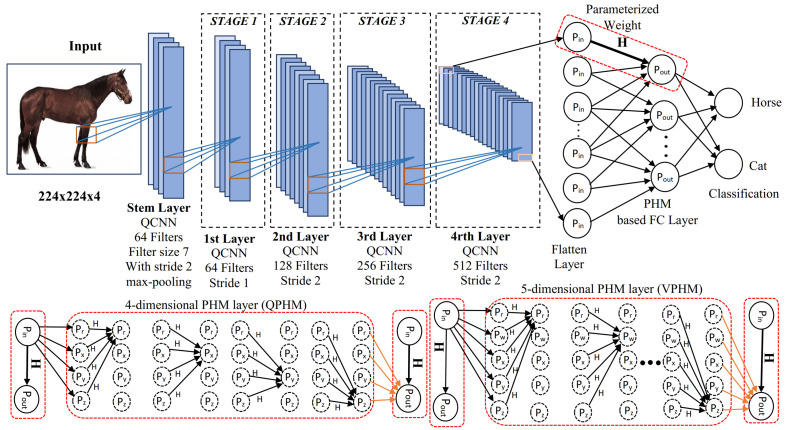
Quaternion network [5] with PHM-based fully connected backend layer (QPHM). Quaternion CNNs are used before the flattened layer to construct the network stages. The bottom row shows the detailed structure of the red rectangle at the top.

**Figure 8 sensors-25-07663-f008:**
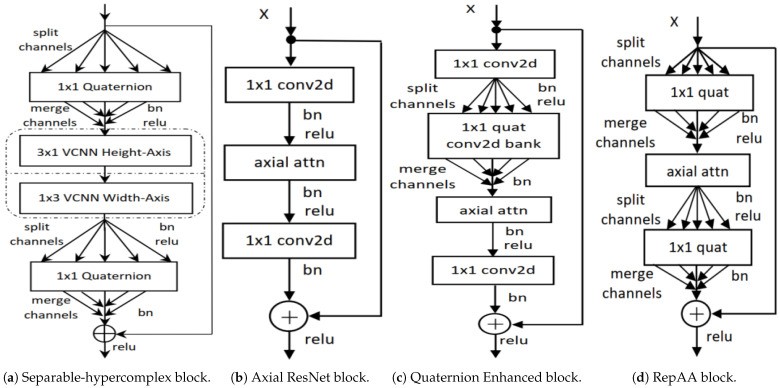
Bottleneck types. “bn”, “attn”, “VCNN”, and “quat” stand for batch normalization, attention, vectormap CNN, and quaternion, respectively. (**a**) Original bottleneck modules found in our novel separable-hypercomplex networks [22]. (**b**) Original bottleneck modules found in Axial-ResNet [11], respectively. (**c**) Quaternion-enhanced (QuatE) axial-ResNet bottleneck block found in [18]. (**d**) Quaternion-based representational Axial-ResNet bottleneck module (RepAA) found in [18].

**Figure 9 sensors-25-07663-f009:**
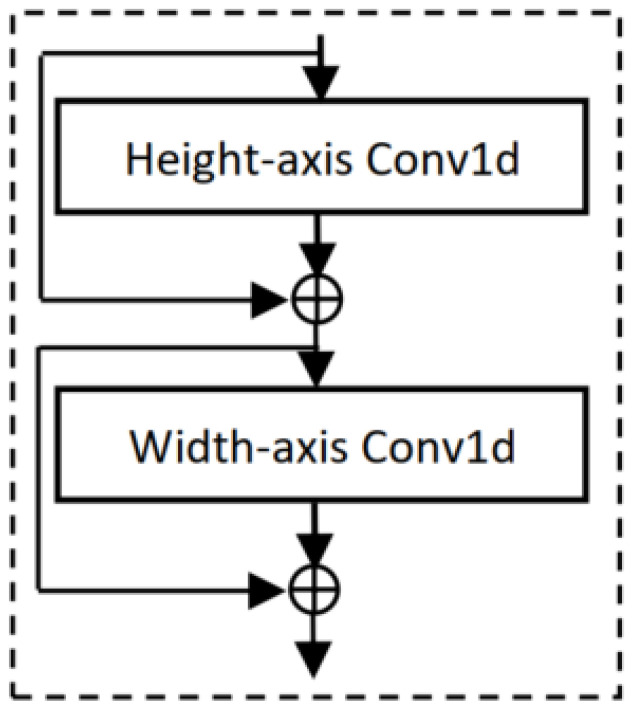
Residual 1D convolutional network (RCN) block used in [19].

**Figure 10 sensors-25-07663-f010:**
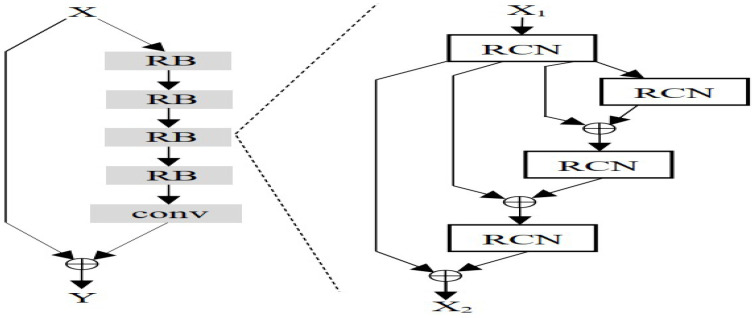
Recursive 1D convolutional residual network (RCRN) architecture with B=4 and U=3 [19]. Here, the “RB” layer refers to a recursive block.

**Figure 11 sensors-25-07663-f011:**
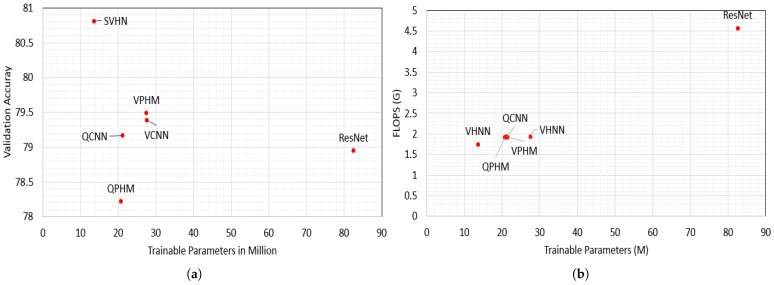
Parameter efficiency analysis of 50 layers ResNet [3], QCNN and VCNN [4], QPHM and VPHM [5], SHNN [22] for the CIFAR100 dataset. (**a**) Validation accuracy versus parameter count. (**b**) FLOPS versus parameter count.

**Figure 12 sensors-25-07663-f012:**
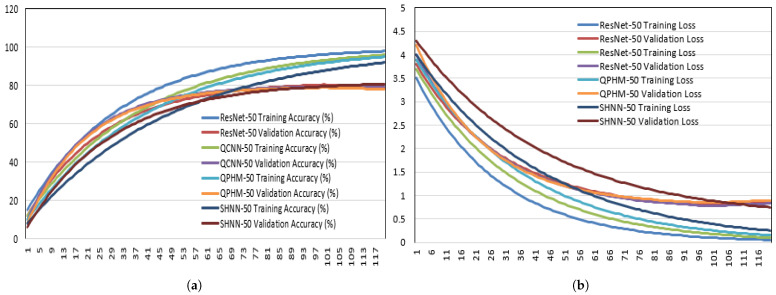
Loss and accuracy curve analysis of 50 layers ResNet [3], QCNN [4], QPHM [5], SHNN [22] for CIFAR100 dataset. (**a**) Accuracy curves. (**b**) Loss curves.

**Table 1 sensors-25-07663-t001:** Image classification performance on CIFAR, SVHN, and tiny-ImageNet datasets. Here, QPHM, VPHM, SHNN, RPHM, and ImageNet define the quaternion networks with the PHM FC layer, vectormap networks with the PHM FC layer, separable hypercomplex networks, ResNets with QPHM backend, and tiny-ImageNet dataset, respectively. Parameters, FLOPS, and latency are calculated for the CIFAR-100 dataset.

Architectures	Params	Flops	Latency	Validation Accuracy			
				CIFAR-10	CIFAR-100	SVHN	ImageNet
ResNet-26 [3]	40.9M	2.56G	0.86 ms	94.68	78.21	96.04	57.21
RPHM-26 [5]	40.8M	2.55G	0.64 ms	95.32	79.14	96.64	57.84
QCNN-26 [9]	10.2M	1.11G	0.65 ms	94.89	77.65	95.88	53.84
VCNN-26 [4]	13.6M	1.09G	0.65 ms	94.76	77.65	95.93	56.15
QPHM-26 [5]	10.2M	1.10G	0.64 ms	95.26	78.15	95.97	54.02
VPHM-26 [5]	13.6M	1.08G	0.67 ms	95.15	78.14	96.24	53.11
EfficientNetV2-B0 [44]	7.10M	0.7B	3.00 ms	95.63	79.28	-	-
SHNN-26 [22]	6.2M	1.06G	0.68 ms	95.91	79.42	97.05	58.56
ResNet-35 [3]	57.8M	3.31G	1.08 ms	94.95	78.72	95.74	57.80
RPHM-35 [5]	57.7M	3.31G	0.81 ms	95.80	79.65	96.22	59.0
QCNN-35 [9]	14.5M	1.47G	0.82 ms	95.33	78.96	95.95	54.53
VCNN-35 [4]	19.3M	1.45G	0.84 ms	95.06	79.52	95.97	55.99
QPHM-35 [5]	14.5M	1.46G	0.79 ms	95.55	78.46	95.99	56.42
VPHM-35 [5]	19.3M	1.44G	0.82 ms	95.60	79.86	96.34	56.10
SHNN-35 [22]	9.2M	1.36G	0.84 ms	96.49	80.09	96.90	60.06
ResNet-50 [3]	82.5M	4.57G	1.32 ms	94.08	78.95	95.76	59.06
RPHM-50 [5]	82.5M	4.57G	1.21 ms	95.86	79.89	96.78	60.30
QCNN-50 [9]	21.1M	1.93G	1.06 ms	95.42	79.17	96.24	56.63
VCNN-50 [4]	27.6M	1.93G	1.13 ms	95.37	79.39	96.39	57.52
QPHM-50 [5]	20.7M	1.92G	1.06 ms	95.75	78.22	96.46	59.42
VPHM-50 [5]	27.5M	1.92G	1.08 ms	95.76	79.49	96.49	58.96
SHNN-50 [22]	13.6M	1.75G	1.09 ms	96.74	80.81	97.25	62.44

**Table 2 sensors-25-07663-t002:** Top-1 validation accuracy for hypercomplex networks. “DCN” stands for deep complex CNN. * variant used quaternion batch normalization.

Models	Accuracy	
	Cifar10	Cifar100
DCNs [16]	38.90	42.6
DCN [15]	94.53	73.37
QCNN [45]	77.48	47.46
Quat [46]	77.78	-
QCNN [35]	83.09	-
QCNN * [35]	84.15	-
Quaternion18	94.80	71.23
Quaternion34	94.27	72.76
Quaternion50	93.90	72.68
Octonion [14]	94.65	75.40
Vectormap18	93.95	72.82
Vectormap34	94.45	74.12
Vectormap50	94.28	74.84
QPHM-50	95.59	80.25
VPHM-50	95.48	78.91
SHNN-50	96.74	80.81

**Table 3 sensors-25-07663-t003:** Classification performance on the ImageNet300k dataset. Here, “QuatE-1”, “QuatE-2”, and “Accur” stand for QuatE axial-attention, QuatE axial-attention with QPHM, and top-1 validation accuracy, respectively.

Models	Params	Accuracy
ResNet-26 [3]	13.6M	45.48
QCNN-26 [9]	15.1M	50.09
QPHM-26 [5]	11.4M	52.23
Axial attention-26 [11]	5.7M	54.79
RepAA-26 [18]	3.7M	60.70
ResNet-35 [3]	18.5M	48.99
QCNN-35 [9]	20.5M	48.11
QPHM-35 [5]	17.5M	51.84
Axial attention-35 [11]	8.4M	60.49
QuatE-1-33 [18]	6.0M	62.30
QuatE-2-33 [18]	5.3M	61.27
RepAA-35 [18]	4.6M	62.03
ResNet-50 [3]	25.5M	50.92
QCNN-50 [9]	27.6M	49.69
QPHM-50 [5]	24.5M	54.38
Axial attention-50 [11]	11.5M	55.7
QuatE-1-66 [18]	11.9M	59.71
QuatE-2-66 [18]	11.1M	62.46
RepAA-50 [18]	6.7M	62.49

**Table 4 sensors-25-07663-t004:** Image classification performance on the CIFAR benchmarks, SVHN, and Tiny ImageNet datasets for 26, 35, 50, 101, and 152-layer ResNet architectures, MobileNets, SqueezeNext with widening factor 1, and SqueezeNext with widening factor 2. The parameters, FLOPS, and latency are analyzed for CIFAR benchmark datasets. These will be a little bit higher for SVHN and ImageNet datasets.

Architectures	Params	Flops	Latency	Validation Accuracy			
				CIFAR10	CIFAR100	SVHN	ImageNet
EfficientNetV2-B0 [44]	7.10M	86.4M	3.00 ms	95.63	79.28	-	-
RCN-EfficientNetV2-B0	6.02M	58.7M	2.04 ms	96.41	82.95	-	-
ResNet-26 [3]	40.9M	2.56G	0.86 ms	94.68	78.21	96.04	57.21
RCN-26 [19]	9.4M	0.68G	0.52 ms	96.08	79.66	97.83	62.28
ResNet-35 [3]	58.1M	3.26G	0.96 ms	94.95	78.72	95.74	57.80
RCN-35 [19]	13.4M	0.86G	0.60 ms	96.15	80.38	97.50	59.31
ResNet-50 [3]	82.5M	4.58G	1.11 ms	95.08	78.95	95.76	59.06
RCN-50 [19]	19.2M	1.16G	0.73 ms	96.25	81.29	97.32	62.40
ResNet-101 [3]	149M	8.8G	1.68 ms	95.36	78.80	96.29	60.62
RCN-101 [19]	34.8M	2.18G	1.28 ms	96.27	80.88	97.29	64.18
ResNet-152 [3]	204M	13.1G	2.36 ms	95.36	79.85	96.35	61.57
RCN-152 [19]	48M	3.2G	1.80 ms	96.37	80.94	97.38	66.16
MobileNet [21]	3.2M	12M	0.18 ms	87.87	60.64	94.23	-
RCMobNet [19]	0.8M	4M	0.18 ms	93.34	61.1	94.53	-
SqNxt-1 [12]	0.6M	59M	0.55 ms	92.30	69.70	95.88	-
RCSqNxt-1 [19]	0.4M	45M	0.37 ms	93.34	70.14	97.13	-
SqNxt-2 [12]	2.3M	226M	0.78 ms	93.38	73.05	96.06	-
RCSqNxt-2 [19]	1.7M	168M	0.47 ms	94.91	74.94	97.40	-

**Table 5 sensors-25-07663-t005:** Image classification performance on the CIFAR benchmarks for 26-layer architectures with different widening factors.

Model Name	Top-1 Validation Accuracy	
	CIFAR-10	CIFAR-100
WRN-28-10 [1]	94.68	79.57
WRCN-26-2 [19]	96.32	83.54
WRCN-26-4 [19]	96.68	83.75
WRCN-26-6 [19]	96.77	83.78
WRCN-26-8 [19]	96.83	83.82
WRCN-26-10 [19]	96.87	83.92

**Table 6 sensors-25-07663-t006:** Benchmark testing PSNR results for scaling factors ×2, ×3, and ×4 on Set5 dataset.

Model Architecture	Scale		
	×2	×3	×4
SRCNN [23]	36.66	32.75	30.48
VDSR [25]	37.53	33.66	31.35
DRCN [24]	37.63	33.82	31.53
DRRN19 [26]	37.66	33.93	31.58
DRRN125 [26]	37.74	34.03	31.68
RCRN19 [19]	37.73	33.99	31.63
RCRN25 [19]	37.84	34.11	31.84

**Table 7 sensors-25-07663-t007:** Ablation study of kernel size *k* for the RCMobileNet on CIFAR-10 and CIFAR-100.

*k*	Parameters (M)	CIFAR-10 (%)	CIFAR-100 (%)
3	0.62	92.84	60.51
5	0.78	93.67	61.18
7	0.94	94.21	61.63
9	1.11	94.28	62.71
11	1.28	94.10	63.55

**Table 8 sensors-25-07663-t008:** Ablation results for kernel size *k* in RCSqueezeNext on CIFAR-10 and CIFAR-100.

*k*	Parameters (M)	CIFAR-10 (%)	CIFAR-100 (%)
3	0.39	93.34	70.14
5	0.54	93.77	70.74
7	0.70	94.91	71.01
9	0.85	93.77	70.79
11	0.99	93.39	70.79

## Data Availability

The data presented in this study are openly available in kaggle https://www.kaggle.com/.

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
