# Peer review of "Analyzing Parameter-Efficient Convolutional Neural Network Architectures for Visual Classification"

_sensors, 2025, doi:10.3390/s25247663_

Round 1
Reviewer 1 Report
Comments and Suggestions for Authors
This paper sorts out the key pathways in the design of parameter-efficient CNNs in recent years. The following improvements still need to be made:
1. It is recommended to supplement the comparison between hypercomplex networks (such as SHNN) and emerging parameter-efficient models in recent years (such as EfficientNetV2) to clarify their performance positioning and differences.
2. The basis for selecting the size of the 1D convolution kernel in RCN, as well as the impact of different kernel sizes on classification accuracy and parameter count, need to be elaborated in detail.
3. The performance of each model in few-shot visual classification tasks should be supplemented to improve the analysis of applicable scenarios for parameter-efficient models.
4. It is suggested to add experiments related to the training stability of hypercomplex networks (FHNN, SHNN), such as the comparison of loss convergence curves.
5. It is suggested that the authors cite the latest literature, as there are no relevant references from 2025 in the paper.
Reviewer 2 Report
Comments and Suggestions for Authors
1. Lack of Authoritativeness and Novel Contribution
There isn't a strong sense of ownership or authority for the suggested work in the paper. The authors mostly provide summaries of earlier research without exhibiting a deep level of analysis or a fresh experimental contribution. Instead of being an original research article, a large portion of the content reads like an extended literature review. The contribution and methodology statements are ambiguous and don't explain what is novel, why it matters, or how it enhances current models.
2. Unclear Objectives and Research Scope
The paper's goals are neither explicitly stated nor rationally related to those of the other sections. The research questions and hypotheses are difficult for the reader to understand. Additionally, it's unclear from the contribution claims whether the work presents a novel model, suggests a change, or just examines CNN architectures that already exist.
3. Poor Organization and Flow
The structure of the manuscript is not coherent or logical. The narrative does not lead the reader through the problem definition, suggested approach, experimentation, and results, and section transitions are abrupt. While the experimental or analytical sections are underdeveloped, many of the subsections (e.g., 2.1, 2.2, 2.3) are repetitive and excessively descriptive. This disparity reduces the paper's academic clarity and makes it challenging to read.
4. Overreliance on Citations — Reads Like a Survey
Rather than being a technical research article, the manuscript reads more like a survey or review. For instance:
About ten references are made in a row in the Introduction's opening paragraph, which makes it difficult to read and raises questions about the original context-setting.
Similarly, it is challenging to discern the author's interpretation from the cited materials in Section 2.1.1 because there are thirteen references in a single paragraph.
Concerns concerning whether the authors critically examined or merely compiled the body of existing literature are also raised by this excessive citation clustering.
Readability is adversely affected by a number of formatting problems:
With its erratic paragraph alignment and ambiguous figure placement, page 2 seems haphazard.
5. Formatting and Presentation Issues
Figure 2 is not clear or professional; it looks more like a rough sketch than an academic figure. Interpretation is challenging due to its geometric layout and the equations next to it (with mismatched parentheses).
The formatting of the equations is inconsistent; for example, Equation (6) and the equations that follow are not appropriately referenced or contextualized.
6. Poor Technical and Mathematical Support
Equations are given without sufficient justification or derivation. For instance:
Without defining every variable or providing evidence for its derivation, equation (6) is introduced suddenly.
Subsequent equations are reiterated with minor modifications, but their contributions to model design or analysis are not explained.
There is no verifiable evidence (algorithms, experiments, or datasets) to back up the authors' claims that they "propose" models like FHNNs or SHNNs.
8. Figures, Equations, and References
-
Figures are unclear, unnumbered in sequence, or inadequately captioned.
-
Several equations are copied without contextual interpretation.
-
References are inconsistently formatted, with overlapping or repeated citations.
Reviewer 3 Report
Comments and Suggestions for Authors
- Clarify and highlight the paper’s contributions.
I recommend revising the introduction to explicitly list and briefly describe the novel contributions, FHNNs, RepAA, SHNNs, and the 1D RCN architecture. Presenting these up front will make the paper’s originality easier to grasp. - Expand the experimental evaluation.
The empirical results would be more compelling if the proposed models were tested on larger and more diverse datasets. This would strengthen the evidence that the efficiency gains generalize beyond small-scale benchmarks. - Improve organization and readability of the related-work section.
The related-work portion is lengthy and highly technical. Reorganizing it into smaller, clearly defined subsections with short summaries would greatly improve clarity and help readers navigate the prior research landscape.
Round 2
Reviewer 1 Report
Comments and Suggestions for Authors
Accept
Author Response
We are grateful for the reviewer’s time, effort, and valuable insights. Thanks for accepting our paper.
Reviewer 2 Report
Comments and Suggestions for Authors
Table 1: Change the location of it It's not clear to insert it at the introduction section
+++++++++++++++++++++++++++++++++++
Again, the author insists on multisection with short description for it for example,
Applications of Wide Residual Networks
Complexity Analysis
Applications of MobileNet Architectures
+++++++++++++++++++++++++++++++++++++
The Complexity Analysis section
The author again inserts equations without numbering
+++++++++++++++++++++++++++
Table 3. Top-1 validation accuracy for hypercomplex networks. “DCN” stands for deep complex CNN. * variant used quaternion batch normalization.
in the middle of the section
++++++++++++++++++++++++++++++++++++++
The caption of the table is suppose to be above the table
++++++++++++++++++++++++++++++++++
do you think this statment needs 5 references
While developing hypercomplex CNNs (HCNNs), it was found that they share weights across channels as well as space [4, 5, 7–9, 14].
++++++++++++++++++++++++++++++
equation 20 make less ident so it will be clear
++++++++++++++++++++++++++++
Author Response
The manuscript has been revised based on the reviewer’s constructive feedback. Table 1 has been relocated from the Introduction to the end of Section 2 to improve readability, and application-related subsections have been removed to enhance structure and coherence, while the Complexity Analysis section has been retained as it is a core component of the work. Table captions, including that of Table 3, have been repositioned above their respective tables to follow standard formatting guidelines. Additionally, the number of references supporting the HCNN statement has been reduced for clarity, and Equation 20 has been reformatted with less indentation to improve visual clarity. Please check the attached file for more details.

Round 3
Reviewer 2 Report
Comments and Suggestions for Authors
The authors have satisfactorily addressed all reviewer comments and significantly improved the overall quality of the manuscript. The revisions enhance both clarity and technical contribution. Therefore, I recommend the paper for publication.